# A Hierarchical Probabilistic Framework for Incremental Knowledge Tracing in Classroom Settings

## Abstract

Knowledge tracing (KT) aims to estimate a student's evolving knowledge state and predict their performance on new exercises based on performance history. Many realistic classroom settings for KT are typically low-resource in data and require online updates as students' exercise history grows, which creates significant challenges for existing KT approaches. To restore strong performance under low-resource conditions, we revisit the hierarchical knowledge concept (KC) information, which is typically available in many classroom settings and can provide strong prior when data are sparse. We therefore propose Knowledge-Tree-based Knowledge Tracing ($KT^2$), a probabilistic KT framework that models student understanding over a tree-structured hierarchy of knowledge concepts using a Hidden Markov Tree Model. $KT^2$ estimates student mastery via an EM algorithm and supports personalized prediction through an incremental update mechanism as new responses arrive. Our experiments show that $KT^2$ consistently outperforms strong baselines in simulated online, low-resource settings.

## 1 Introduction

Knowledge tracing (KT) refers to an educational task that, given historical exercises performance of a set of students, aims to predict whether they can solve a new exercise correctly. It can be regarded as modeling the evolution of a student's knowledge during learning, which is essential in personalized learning systems, enabling dynamic adaptation to students' needs. While prior works on knowledge tracing impose very loose constraints about the availability of data (Piech et al., 2015; Ghosh et al., 2020), in many practical real-world classroom scenarios, these constraints tend to be much more stringent, due to various privacy and usability considerations. In particular, three practical constraints have usually been understudied.

• **Cold Start.** For each target student, existing works usually assume that abundant historical data from that student is available before the prediction starts. This implies that the prediction would need to happen at a very late stage of the student's learning – only after many exercises have been done. To make KT meaningful, the prediction should start shortly after the target student starts learning, with only a few exercises completed. This creates a cold start scenario for KT.

• **Online Update.** Conventional KT methods keep their model parameters fixed once the training phase is completed, and apply the same parameters to predict all test data. However, in practice, new student exercises constantly arrive in a streaming fashion, requiring KT methods to be efficiently updated to capture the evolution of students' understanding.

• **Limited Peers.** Many KT methods require a large number of peers to form a large dataset as the training set. In practice, the number of peers may be limited due to privacy protection.

The aforementioned constraints create significant challenges for existing KT algorithms. Specifically, recent KT approaches can be broadly categorized into two types. The first type, *the deep-learning-based methods* (Zhang et al., 2017; Ghosh et al., 2020; Pandey & Karypis, 2019; Liu et al., 2023b), trains neural architectures to model complex student performance patterns. Although these methods have shown strong performance

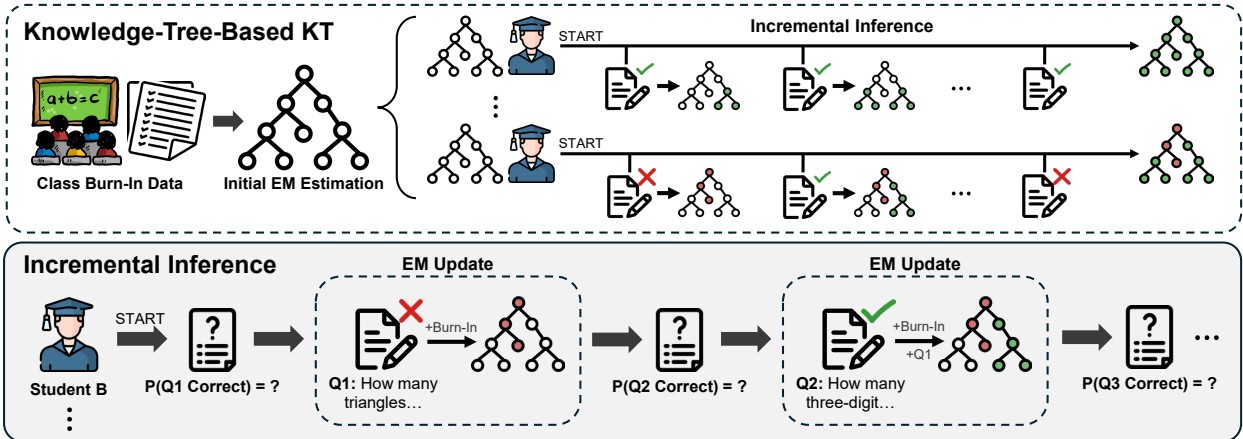

Figure 1: $KT^2$ framework. The model first performs a class-level estimation using burn-in responses to initialize a global knowledge model. Each student is assigned a personalized knowledge concept tree, which is incrementally updated with each new interaction via one-step EM.

under benchmark settings, they require large amounts of training data to achieve good performance, which is unavailable under the practical constraints.

On the other hand, the second type of methods, the *large language model (LLM)-based methods* (Li et al., 2024; Neshaei et al., 2024), leverages LLMs to perform the prediction tasks, either fine-tuning the LLM or selecting a subset of the target student's historical data as in-context examples. While these methods have improved data efficiency and online flexibility, they often struggle to capture fine-grained patterns in historical data and may make predictions based on naive decision rules.

In summary, there appears to be a fundamental trade-off between data availability and a model's capacity to capture complex dependencies in student responses across exercises. Yet, one crucial source of information—long underused in existing KT methods—could help overcome this bottleneck: **knowledge concepts**.

Knowledge concepts (KCs) refer to the labeled topics, knowledge areas, or skills associated with each exercise. These concepts often follow a hierarchical structure, where broader concepts encompass finer-grained sub-concepts. KCs are readily available in many classroom settings, and can be effective for modeling student learning. Intuitively, if a student has not mastered a particular concept, they are likely to answer related questions incorrectly. Moreover, their performance on conceptually adjacent or dependent concepts may also be affected. In low-resource classroom settings, these knowledge labels, along with their hierarchical organization, can provide strong structural priors that enhance prediction performance. **Can we leverage hierarchical knowledge concepts to enable data-efficient, flexible, and online knowledge tracing for low-resource classroom settings?**

Motivated by this observation, we propose Knowledge-Tree-based Knowledge Tracing ($KT^2$), a probabilistic framework for low-resource, online knowledge tracing. $KT^2$ builds a Hidden Markov Tree Model (Crouse et al., 2002) based on the hierarchical tree structure of KCs, where the hidden variables represent the student's mastery of each KC, and the observed variables correspond to their correctness on individual exercise questions. As shown in Fig. 1, the model parameters are first estimated using a small amount of initial ("burn-in") data via the standard Expectation-Maximization (EM) algorithm (Dempster et al., 1977), and are then incrementally updated as new exercise responses are observed. $KT^2$ enables principled prediction of future performance of target students by computing the distribution of correctness on upcoming exercises variables, conditioned on the students' historical exercise performance.

Our experiments on simulated low-resource testeds demonstrate that $KT^2$ effectively addresses the challenges of low-resource knowledge tracing. For instance, with only 100 target students and as few as ten exercises per student, $KT^2$ consistently outperforms the strongest baselines. In the online setting, $KT^2$ requires only a single EM step to incorporate new data, enabling efficient and consistent performance updates.

## 2  Related Works

**Traditional Machine Learning KT** The earliest KT models are based on probabilistic or statistical machine learning methods. BKT (Corbett & Anderson, 1994) formulates the task as a Hidden Markov Model over students' latent knowledge states. Extensions such as PFA (Pavlik Jr et al., 2009) employ logistic regression to model the effect of practice and forgetting, while IRT (Cai et al., 2016) focuses on estimating student ability and item difficulty. Later works, such as iBKT (Yudelson et al., 2013), adapt these methods to incorporate personalization. More recent work also studies cold-start mitigation by introducing evolutionary features based on decayed student-level and KC-level performance histories (Eglington et al., 2025).

**Deep Learning KT** In recent years, with the rapid development of deep learning techniques, many knowledge tracing methods have adopted neural models to predict student performance. Early models include RNN-based DKT (Piech et al., 2015), memory-augmented DKVMN (Zhang et al., 2017), and its IRT-inspired neural variant Deep-IRT (Yeung, 2019). EKT (Liu et al., 2019) further incorporates exercise content and knowledge concept effects to support both student performance prediction and analysis of knowledge acquisition. Later works such as SAKT (Pandey & Karypis, 2019) and GIKT (Yang et al., 2021) adopt attention mechanisms to better model the importance of historical exercises. Transformer-based KT models, such as AKT (Ghosh et al., 2020), qDKT (Sonkar et al., 2020), SAINT (Choi et al., 2020), and AT-DKT (Liu et al., 2023a), further improve this by capturing long-range dependencies and contextual signals. ReKT (Shen et al., 2024) adopts architecture inspired by human cognitive development theories with only two linear units, achieve strong performance with simple designs. Recent KT methods further study how to represent student knowledge states more transparently. for example, MIKT (Sun et al., 2024) traces students' knowledge states at both domain and concept levels, and introduces an IRT-based prediction module to improve interpretability. Despite these improvements, many deep KT models still treat KCs as independent and represent student knowledge in latent vectors, limiting interpretability and generalizability.

**Structure-Aware KT** Several works have attempted to incorporate structural relationships among KCs. Prerequisite-driven methods such as PDKT (Chen et al., 2018) explicitly incorporate prerequisite relations among concepts to guide knowledge tracing. Recent graph-based KT models such as GKT (Nakagawa et al., 2019), DHKT (Wang et al., 2019), SKT (Tong et al., 2020), GIKT (Yang et al., 2021), SHDKT (Yang et al., 2022), KSGAN (Mao et al., 2022), and HKT (Li et al., 2025) enhance embeddings by encoding concept co-occurrence or hierarchy via GNNs or attention. More recent approaches like HHSKT (Ni et al., 2023) construct heterogeneous graphs over learner-question interactions with hierarchical differentiation, while PSI-KT (Zhou et al., 2024) jointly models learner traits and prerequisite structure using a hierarchical Bayesian generative model. These models primarily treat structure as auxiliary input or require substantial learner histories for inference.

**LLM-Based KT** LLMs have seen remarkable progress in recent years (Achiam et al., 2023; Anthropic, 2025; DeepMind, 2025), leading to growing interest in their application in education, including online tutoring, feedback generation, and question generation (Heffernan et al., 2024; Liu et al., 2024; Luo et al., 2024). Recent works have explored applying LLMs for KT, aiming to improve interpretability and reduce data dependence. One line of work adopts few-shot prompting to estimate student mastery and generate natural language rationales (Li et al., 2024). These methods require repeated prompting, are sensitive to example selection, and lead to high computational and financial costs. Another line of work investigates fine-tuning LLMs for KT. Current results show that fine-tuned LLMs can match classical probabilistic models like BKT, but they still lag behind modern deep KT models' performance (Neshaei et al., 2024; Lee et al., 2024). Recent LLM-based KT methods further target cold-start settings by using LLM-estimated question attributes for cold-start prediction (Guo et al., 2024) or reformulating student interaction histories as natural language to fine-tune a generative LLM as a knowledge tracer (Jung et al., 2025). Another recent direction uses LLM-based teacher and student agents to model hierarchical knowledge dependencies and aligns synthetic and real behavioral patterns in hyperbolic space for knowledge tracing (Fu et al., 2026). While LLMs offer flexibility and reasoning capabilities, further adaptation is needed for real-world deployment.

## 3    Methods

### 3.1    Problem Formulation

The problem of KT can be formulated as follows. Denote $\mathcal{I}$ as a set of student IDs. For each $i \in \mathcal{I}$, denote $\mathcal{N}_i$ as a set of exercise IDs for which student $i$ has solved. For each $i \in \mathcal{I}$, and $n \in \mathcal{N}_i$, define $Q_{ni}$ as a random binary variable:

$$Q_{ni} = \begin{cases} 1 & \text{if student } i \text{ answers question } n \text{ correctly,} \\ 0 & \text{otherwise.} \end{cases} \tag{1}$$

Finally, define $\mathcal{Q}_i = \{Q_{ni} : n \in \mathcal{N}_i\}$ as a set of historical exercise variables of student $i$. Given a new question $n^*$, our goal is to predict:

$$p_{\boldsymbol{\theta}}(Q_{n^*i} = 1 | \mathcal{Q}_i). \tag{2}$$

Here, $\boldsymbol{\theta}$ represents a set of parameters that defines the distribution, which will be estimated based on all the observed historical data, $\cup_i \mathcal{Q}_i$.

We here restate the real-world constraints mentioned in Sec. 1 formally:

● **Cold Start.** At the onset, each $\mathcal{Q}_i$ is a very small set, making the conditional probability in Eq. equation 2 uninformative and the estimation of $\boldsymbol{\theta}$ challenging.

● **Online Update.** Each $\mathcal{Q}_i$ is expanding, requiring Eq. equation 2 and $\boldsymbol{\theta}$ to be constantly updated.

● **Limited Peers.** $\mathcal{I}$ is a small set, adding to data scarcity when estimating $\boldsymbol{\theta}$.

In the following, we will discuss how the hierarchical KCs can alleviate the challenges.

### 3.2    Knowledge Concept Tree

Many hierarchical KC structures can be organized into trees. Fig. 2(left) shows a portion of an example knowledge concept tree, where parent nodes represent broader KCs and child nodes finer ones. Each edge represents an entailment relationship. We make two assumptions on our classroom setting: ❶ We have access to a knowledge concept tree that contains all the KCs covered in the exercises; and ❷ each exercise question is labeled with *one KC at the leaf node.*

### 3.3    The KT$^2$ Model

KT$^2$ incorporates the knowledge concept tree information into the probabilistic modeling of the $Q_{ni}$ variables. Specifically, denote $\mathcal{C}$ as the set of all KCs. For each $i \in \mathcal{I}$, and $c \in \mathcal{C}$, define $K_{ci}$ as the following random binary variable:

$$K_{ci} = \begin{cases} 1 & \text{if student } i \text{ masters KC } c, \\ 0 & \text{otherwise.} \end{cases} \tag{3}$$

KT$^2$ introduces a Hidden Markov Tree Model to model the joint probability distribution of $\{Q_{ni}\}$ (as observed variables) and $\{K_{ci}\}$ (as hidden variables) via the following assumptions.

**First**, all the variables are independent across different students, *i.e.*,

$$p\big(\cup_{n,i}\{Q_{ni}\}, \cup_{c,i}\{K_{ci}\}\big) = \prod_i p\big(\cup_n \{Q_{ni}\}, \cup_c \{K_{ci}\}\big). \tag{4}$$

**Second**, for each student $i$, the corresponding probability distribution can be decomposed as

$$p\big(\cup_n \{Q_{ni}\}, \cup_c \{K_{ci}\}\big) = \underbrace{p\big(\cup_c \{K_{ci}\}\big)}_{\text{transition}} \cdot \underbrace{p\big(\cup_n \{Q_{ni}\} | \cup_c \{K_{ci}\}\big)}_{\text{emission}}, \tag{5}$$

where the first term, representing the *transition probabilities*, models interdependencies among the mastery of different KCs; the second term, representing the *emission probabilities*, models how the mastery of KCs indicates the exercise correctness.

**Third (Transition Probabilities)**, for each student $i$, the probability graphical model for $\{K_{ci}\}$ (Fig. 2 (right)) follows the same topological structure as the knowledge concept tree (Fig. 2 (left)). Namely, each $K_{ci}$ only directly depends on the mastery variable of its parent KC:

$$p\big( \cup_c \{K_{ci}\} \big) = \prod_c p(K_{ci}|K_{\mathcal{P}(c)i}), \tag{6}$$

where $\mathcal{P}(c)$ denotes the parent KC of $c$. To model transition probability, $p(K_{ci}|K_{\mathcal{P}(c)i})$, we assume that mastering a parent KC would entail mastering ALL its children (but not vise versa), hence

$$p(K_{ci} = 1 \mid K_{\mathcal{P}(c)i} = k) = \begin{cases} 1 & \text{if } k = 1, \\ \gamma_c & \text{otherwise,} \end{cases} \tag{7}$$

where each $\gamma_c$ is a parameter to be estimated. When $c$ is the root node, Eq. equation 7 still applies with the condition $K_{\mathcal{P}(c)i} = k$ removed.

**Fourth (Emission Probabilities)**, as shown in Fig. 2 (right), we assume each $Q_{ni}$ only directly depends on the mastery variable of its corresponding labeled concepts, i.e.,

$$p\big( \cup_n \{Q_{ni}\}| \cup_c \{K_{ci}\} \big) = \prod_n p(Q_{ni}|K_{\mathcal{M}(n)i}), \tag{8}$$

where $\mathcal{M}(n)$ denotes KC for exercise $n$. Emission probability, $p(Q_{ni}|K_{\mathcal{M}(n)i})$, is modeled as

$$p(Q_{ni} = 1 \mid K_{\mathcal{M}(n)i} = k) = \begin{cases} \phi_n, & \text{if } k = 1, \\ \varepsilon, & \text{otherwise,} \end{cases} \tag{9}$$

where $\phi_n$ represents the correct probability if the student knows the underlying concepts. It has three possible values, $\{r_{easy}, r_{med}, r_{hard}\}$, depending on the difficulty level of the exercise, which is determined by which of the three pre-defined bins (high, medium, or low) the historical solve rate falls into. $\varepsilon$ represents the correct probability by random guessing. $\varepsilon < r_{hard} < r_{med} < r_{easy}$ are then estimated.

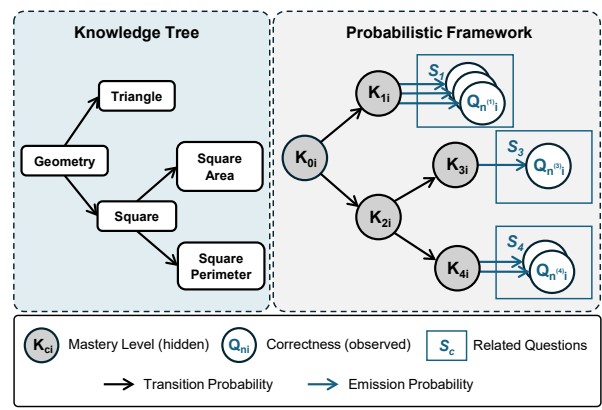

Figure 2: An KC tree structure (left), and its corresponding probabilistic framework (right).

**Summary.** The joint probability of all the hidden and observed variables is defined by Eqs. equation 4 to equation 9. The parameters of the model, $\boldsymbol{\theta}$, include

$$\boldsymbol{\theta} = [\cup_c \gamma_c, r_{easy}, r_{med}, r_{hard}, \varepsilon]. \tag{10}$$

Sec. 3.4 will discuss how to estimate $\boldsymbol{\theta}$. Sec. 3.5 will discuss how to predict the correctness probability (Eq. equation 2).

## 3.4 Parameter Estimation

The parameters $\boldsymbol{\theta}$ are estimated via the maximum log-likelihood objective on the *observed variables*:

$$\max_{\boldsymbol{\theta}} \log p_{\boldsymbol{\theta}}(\mathcal{Q}), \tag{11}$$

where $\mathcal{Q}$ denotes a set of observed historical correctness; $p_{\boldsymbol{\theta}}(\mathcal{Q})$ can be computed by marginalizing Eq. equation 4 over all the hidden variables, $\{K_{ci}\}$ (we add a subscript $\boldsymbol{\theta}$ under $p$ to emphasize that $p$ is parameterized by $\boldsymbol{\theta}$).

Directly computing Eq. equation 11 is computationally expensive due to the marginalization of the hidden variables. We thus adopt the standard EM algorithm (Dempster et al., 1977) for the optimization. EM

algorithm is an iterative algorithm. Denote $\boldsymbol{\theta}^{(\tau-1)}$ as the parameter estimate after iteration $\tau - 1$. Then $\boldsymbol{\theta}^{(\tau)}$ can be computed from $\boldsymbol{\theta}^{(\tau-1)}$ via the objective:

$$\boldsymbol{\theta}^{(\tau)} = \arg\max_{\boldsymbol{\theta}} A(\boldsymbol{\theta}; \boldsymbol{\theta}^{(\tau-1)}), \tag{12}$$

where $A(\boldsymbol{\theta}; \boldsymbol{\theta}^{(\tau-1)})$ equals

$$\mathbb{E}_{p_{\boldsymbol{\theta}^{(\tau-1)}}(\cup_{c,i}\{K_{ci}\}|\mathcal{Q})}[\log p_{\boldsymbol{\theta}}(\cup_{n,i}\{K_{ci}\}, \mathcal{Q})]. \tag{13}$$

It can be shown that EM algorithm converges to the optimal solution to Eq. equation 11. Eq. equation 12 bears a closed-form solution that can be computed efficiently. See Appendix A for details.

### 3.5 Inference

Once $\boldsymbol{\theta}$ is estimated, we can predict if student $i$ can answer question $n^*$ correctly by computing Eq. equation 2, which can be further decomposed into

$$\begin{aligned}
&p_{\boldsymbol{\theta}}(Q_{n^*i} = 1|\mathcal{Q}_i) \\
&= \sum_{k=0}^{1} p_{\boldsymbol{\theta}}(K_{\mathcal{M}(n^*)i} = k|\mathcal{Q}_i) p(Q_{n^*i} = 1|K_{\mathcal{M}(n^*)i} = k, \mathcal{Q}_i) \\
&= p_{\boldsymbol{\theta}}(K_{\mathcal{M}(n^*)i} = 0|\mathcal{Q}_i)\varepsilon + p_{\boldsymbol{\theta}}(K_{\mathcal{M}(n^*)i} = 1|\mathcal{Q}_i)\phi_n,
\end{aligned} \tag{14}$$

where the second equality is derived from the conditional independence assumption in Eq. equation 8; the third from Eq. equation 9.

Eq. equation 14 implies that the prediction process involves two steps. First, infer the KC mastery associated with the target exercise based on the historical performance, *i.e.*, computing $p_{\boldsymbol{\theta}}(K_{\mathcal{M}(n^*)i} = k|\mathcal{Q}_i)$, which can be efficiently computed using the *upward-downward algorithm* (Crouse et al., 2002) (see Appendix B). Second, use the KC mastery probability to modulate the emission probability (last line of Eq. equation 14).

### 3.6 Incremental Update

In online settings, each $\mathcal{Q}_i$ is constantly expanding, making it necessary to re-estimate $\boldsymbol{\theta}$ based on Eqs. equation 11 and equation 12. We design the following incremental update scheme to balance between efficiency and performance, as shown in Fig. 1.

**Communal Burn-in.** At the start, each $\mathcal{Q}_i$ contains only a few data. We aggregate the early data from all target students into a shared burn-in dataset, denoted by $\mathcal{Q}_{init}$, and estimate a common model, $\boldsymbol{\theta}_{init}$, via Eq. equation 11 with $\mathcal{Q} = \mathcal{Q}_{init}$. We run the full EM iterations till convergence. This model is then used to predict the correctness of the first exercise (post-burn-in) for both seen and unseen students.

**Personalized Update.** After the burn-in phase, we maintain personalized models for each target student. Let $\boldsymbol{\theta}_i$ denote the model parameters for student $i$, estimated by Eq. equation 11 with $\mathcal{Q} = \mathcal{Q}_{init} \cup \mathcal{Q}_i$ – that is, the burn-in data from all students and all historical responses from student $i$. As new responses are observed (*i.e.*, as $\mathcal{Q}_i$ expands), we incrementally update $\boldsymbol{\theta}_i$ by performing a single EM iteration, enabling efficient adaptation and personalization.

## 4 Experiment

We first describe the construction of classroom testbeds and baseline settings in Secs. 4.1 and 4.2, respectively. We then compare KT$^2$ with existing baselines and provide analysis in Sec. 4.3 - Sec. 4.5.

### 4.1 Data Construction

We first construct datasets that follow the low-resource scenarios as described in Sec. 1 and that come with knowledge concept trees. We will open-source all our datasets, along with the code for preprocessing and dataset construction.

| XES3G5M | Mean | Std | Interactions |
|---|---|---|---|
| Application Module | 0.6249 | 0.1600 | 10532 |
| Computation Module | 0.6235 | 0.1354 | 7521 |
| Counting Module | 0.6689 | 0.1424 | 7279 |

| MOOCRadar | Mean | Std | Interactions |
|---|---|---|---|
| Wine Knowledge | 0.6592 | 0.1342 | 8469 |
| Circuit Design | 0.6394 | 0.1320 | 17616 |
| Education Theory | 0.6791 | 0.1205 | 8796 |

Table 1: Statistics of simulated classroom data extracted from XES3G5M and MOOCRADAR. Mean and Std correspond to class accuracy rates on all interactions.

**Dataset Selection.** We identify two widely used educational datasets: XES3G5M (Liu et al., 2023c) for K–12 math and MOOCRADAR (Yu et al., 2023) for university-level courses. Each exercise is annotated with fine-grained KCs and can also be assigned a difficulty label (easy/medium/hard) based on student correctness rates. Statistics for the datasets are in Appendix E.

**Knowledge Concept Tree.** Next, we discuss how the knowledge concept trees are structured in these datasets. XES3G5M arranges its KCs into a hierarchical entailment structure, allowing us to use it directly as a knowledge concept tree. For MOOCRADAR, it only provides KCs without any hierarchical structure, so we build a knowledge concept tree for it instead. Inspired by KCQRL (Ozyurt et al., 2024), we encode KCs into semantic embeddings (McInnes et al., 2018) and cluster semantically similar KCs together (Campello et al., 2013). For each cluster, we use GPT-4o-mini (OpenAI, 2024) to generate a representative KC label. Finally, we manually annotate unclustered KCs. Further details are provided in Appendix F.

**Data Sampling.** To simulate the low-resource setting, we need to sample a subset of questions and students. First, to sample a subset of questions and KCs, we partition the knowledge concept trees by taking each level-1 node as the root of a knowledge module. Each module consists of the root and all its descendant KCs. We then identify students who have worked on at least 50 exercises associated with the module. For each dataset, we select the top-3 knowledge modules with the largest number of students satisfying the condition. Then, we remove out-of-module exercises for those students.

We then sample 100 students per module with a good coverage of overall performance. Specifically, we use a normal distribution $\mathcal{N}(0.65, 0.15)$ to draw 100 overall correctness rates. For each drawn correctness rate, we select a student whose overall correctness rate best matches the drawn sample without replacement. Each sampled student's first 10 interactions are added to the burn-in data, and the remaining are used for incremental inference. Table 1 shows the statistics of simulated sets.

## 4.2 Setup

**Metrics.** Following previous works, we evaluate models using AUC, accuracy, and F1 score.

**Baselines.** We consider several commonly used methods, including conventional machine learning methods, deep learning methods, and LLM-based approaches.

• IBKT (Yudelson et al., 2013) is an individualized extension of BKT that assigns separate parameters to each student, modeling student-specific knowledge states with a Hidden Markov Model.

• AKT (Ghosh et al., 2020) is a transformer-based model that uses a monotonic attention mechanism to measure the relevance between current questions and historical exercises.

• SAINT (Choi et al., 2020) is a transformer-based model with a deep encoder-decoder architecture. It separates exercise and response sequences and encodes them respectively, allowing stacked attention layers to learn the dependencies.

• QDKT (Sonkar et al., 2020) is a variant of DKT that models performance at question level. It applies Laplacian regularization to smooth predictions across similar questions and uses fastText initialization to improve generalization.

• SKT (Tong et al., 2020) is a structure-aware method that jointly models the temporal effect of exercises and the spatial effect of multiple concept relations.

| Model | AUC | ACC | F1 | AUC | ACC | F1 | AUC | ACC | F1 | AUC | ACC | F1 |
|---|---|---|---|---|---|---|---|---|---|---|---|---|
| XES3G5M (Liu et al., 2023c) | | | | | | | | | | | | |
| | Application Module | | | Computation Module | | | Counting Module | | | Avg. | | |
| ιBKT (Yudelson et al., 2013) | 0.5630 | 0.6054 | 0.7104 | 0.6011 | 0.6483 | 0.7684 | 0.6093 | 0.7040 | 0.8025 | 0.5911 | 0.6526 | 0.7604 |
| AKT (Ghosh et al., 2020) | 0.6790 | 0.6659 | 0.7418 | 0.6701 | 0.6628 | 0.7560 | 0.6994 | 0.7184 | 0.8114 | 0.6828 | 0.6824 | 0.7697 |
| AKT-Online | 0.7106 | 0.6825 | 0.7645 | 0.6848 | 0.6722 | 0.7690 | 0.7174 | 0.7203 | 0.8080 | 0.7043 | 0.6917 | 0.7805 |
| SAINT (Choi et al., 2020) | 0.6031 | 0.6423 | 0.7822 | 0.6079 | 0.6306 | 0.7735 | 0.5794 | 0.6886 | 0.8156 | 0.5968 | 0.6538 | 0.7904 |
| SAINT-Online | 0.6899 | 0.6826 | 0.7615 | 0.6641 | 0.6609 | 0.7558 | 0.6779 | 0.6904 | 0.7834 | 0.6773 | 0.6780 | 0.7669 |
| qDKT (Sonkar et al., 2020) | 0.5177 | 0.5178 | 0.5965 | 0.5338 | 0.5272 | 0.5897 | 0.4856 | 0.4716 | 0.5410 | 0.5124 | 0.5055 | 0.5757 |
| qDKT-Online | 0.6663 | 0.6358 | 0.7185 | 0.6606 | 0.6363 | 0.7138 | 0.6988 | 0.6697 | 0.7603 | 0.6752 | 0.6473 | 0.7309 |
| SKT (Tong et al., 2020) | 0.5195 | 0.6787 | **0.8062** | 0.5076 | 0.6381 | 0.7738 | 0.5234 | 0.7216 | **0.8369** | 0.5168 | 0.6795 | **0.8056** |
| REKT (Shen et al., 2024) | 0.7068 | 0.6726 | 0.7458 | 0.6909 | 0.6725 | 0.7633 | 0.6847 | 0.7150 | 0.8199 | 0.6941 | 0.6867 | 0.7763 |
| CSKT (Bai et al., 2025a) | 0.6694 | 0.6573 | 0.7411 | 0.6850 | 0.6561 | 0.7667 | 0.6882 | 0.7020 | 0.8127 | 0.6809 | 0.6718 | 0.7735 |
| LEFOKT (Bai et al., 2025b) | 0.6595 | 0.6506 | 0.7343 | 0.6756 | 0.6675 | 0.7670 | 0.5997 | 0.6891 | 0.8160 | 0.6450 | 0.6691 | 0.7724 |
| QWEN-2.5 (Qwen-Team, 2025) | 0.5939 | 0.6425 | 0.7823 | 0.5730 | 0.6301 | 0.7731 | 0.5918 | 0.6857 | 0.8136 | 0.5862 | 0.6528 | 0.7897 |
| + CoT | 0.5534 | 0.6406 | 0.7809 | 0.5746 | 0.6320 | 0.7745 | 0.5919 | 0.6883 | 0.8153 | 0.5733 | 0.6536 | 0.7902 |
| LLAMA-3.2 (Meta, 2024) | 0.5839 | 0.6431 | 0.7825 | 0.5649 | 0.6304 | 0.7732 | 0.5873 | 0.6868 | 0.8136 | 0.5787 | 0.6534 | 0.7898 |
| + CoT | 0.6188 | 0.6440 | 0.7824 | 0.5871 | 0.6329 | 0.7748 | 0.6154 | 0.6918 | 0.8162 | 0.6071 | 0.6562 | 0.7911 |
| **Ours** | **0.7448** | **0.7057** | 0.7962 | **0.7079** | **0.6952** | **0.7807** | **0.7470** | **0.7326** | 0.8258 | **0.7332** | **0.7111** | 0.8009 |
| MoocRadar (Yu et al., 2023) | | | | | | | | | | | | |
| | Wine Knowledge | | | Circuit Design | | | Education Theory | | | Avg. | | |
| ιBKT (Yudelson et al., 2013) | 0.6019 | 0.6719 | 0.7873 | 0.5651 | 0.6331 | 0.7654 | 0.5635 | 0.6794 | 0.8037 | 0.5768 | 0.6615 | 0.7855 |
| AKT (Ghosh et al., 2020) | 0.5901 | 0.6546 | 0.7784 | 0.6142 | 0.6402 | 0.7722 | 0.5535 | 0.6910 | 0.8173 | 0.5859 | 0.6619 | 0.7893 |
| AKT-Online | 0.7208 | 0.6967 | 0.7955 | 0.6726 | 0.6712 | 0.7749 | 0.6442 | 0.6974 | 0.8059 | 0.6792 | 0.6884 | 0.7921 |
| SAINT (Choi et al., 2020) | 0.4905 | 0.6568 | 0.7929 | 0.6066 | 0.6347 | 0.7766 | 0.5918 | 0.6910 | 0.8173 | 0.5630 | 0.6608 | 0.7956 |
| SAINT-Online | 0.5646 | 0.6510 | 0.7767 | 0.6284 | 0.6389 | 0.7554 | 0.6052 | 0.6878 | 0.8085 | 0.5994 | 0.6592 | 0.7802 |
| qDKT (Sonkar et al., 2020) | 0.5857 | 0.5629 | 0.6371 | 0.5248 | 0.5106 | 0.5774 | 0.5420 | 0.5142 | 0.5918 | 0.5508 | 0.5292 | 0.6021 |
| qDKT-Online | 0.7661 | 0.7230 | 0.7924 | 0.7063 | 0.6716 | 0.7438 | 0.7327 | 0.7292 | 0.8118 | 0.7350 | 0.7079 | 0.7827 |
| SKT (Tong et al., 2020) | 0.5095 | 0.6477 | 0.7846 | 0.4908 | 0.6382 | 0.7780 | 0.5099 | 0.6669 | 0.7989 | 0.5034 | 0.6509 | 0.7872 |
| REKT (Shen et al., 2024) | 0.6313 | 0.6248 | 0.7264 | 0.5490 | 0.5254 | 0.6228 | 0.5792 | 0.6896 | 0.8154 | 0.5865 | 0.6133 | 0.7215 |
| CSKT (Bai et al., 2025a) | 0.5849 | 0.6558 | 0.7921 | 0.5454 | 0.5308 | 0.6310 | 0.6317 | 0.6861 | 0.8138 | 0.5873 | 0.6242 | 0.7456 |
| LEFOKT (Bai et al., 2025b) | 0.5620 | 0.6558 | 0.7921 | 0.5497 | 0.5911 | 0.7156 | 0.6252 | 0.6861 | 0.8138 | 0.5790 | 0.6443 | 0.7738 |
| QWEN-2.5 (Qwen-Team, 2025) | 0.6190 | 0.6526 | 0.7898 | 0.5595 | 0.6377 | 0.7788 | 0.5454 | 0.6861 | 0.8138 | 0.5746 | 0.6588 | 0.7941 |
| + CoT | 0.6394 | 0.6831 | 0.7878 | 0.5960 | 0.6340 | 0.7600 | 0.6081 | 0.6830 | 0.8026 | 0.6145 | 0.6670 | 0.7835 |
| LLAMA-3.2 (Meta, 2024) | 0.5889 | 0.6526 | 0.7898 | 0.5852 | 0.6385 | 0.7790 | 0.5562 | 0.6887 | 0.8142 | 0.5768 | 0.6599 | 0.7943 |
| + CoT | 0.6002 | 0.6553 | 0.7899 | 0.5948 | 0.6379 | 0.7789 | 0.5882 | 0.6880 | 0.8141 | 0.5844 | 0.6604 | 0.7943 |
| **Ours** | **0.7714** | **0.7354** | **0.8237** | **0.7662** | **0.7315** | **0.7906** | **0.7891** | **0.7656** | **0.8329** | **0.7756** | **0.7442** | **0.8157** |

Table 2: Knowledge tracing performance comparison on XES3G5M and MoocRadar. **Bold** number indicates the best performance in each module, and underlined number indicates the second best. The Avg. column reports the average result across all modules in each dataset.

- REKT (Shen et al., 2024) is a lightweight model that models student state from question, concept, and domain perspectives by a Forget-Response-Update (FRU) architecture.

- CSKT (Bai et al., 2025a) is a cold-start-oriented model that combines a kernel bias for short-to-long sequence extrapolation with hyperbolic cone attention to capture hierarchical concept relationships.

- LEFOKT (Bai et al., 2025b) is a forgetting-aware framework that enhances attention based KT models with relative forgetting attention, decoupling forgetting patterns from problem relevance via relative positional encodings while improving length extrapolation.

- LLM's ability to handle long-context inputs makes them potentially capable for predicting student performance based on historical exercises. We adopt two different LLM architectures: QWEN-2.5-7B (Qwen-Team, 2025) and LLAMA-3.2-3B (Meta, 2024). During inference, each model is given 10 historical data of the current student and predicts the correctness of the next. We report results under both vanilla prompting and chain-of-thought prompting (Wei et al., 2022).

We also implement an online variant for three deep learning knowledge tracing (DLKT) baselines, denoted as AKT-Online, SAINT-Online, and qDKT-Online. These models are initially trained on the same burn-in data as their offline versions. During inference, we aggregate all students' cumulative interactions into the training set and perform an additional round of training. The updated model is then used to predict the next question for all students. Note that this online strategy gives the baselines access to more training

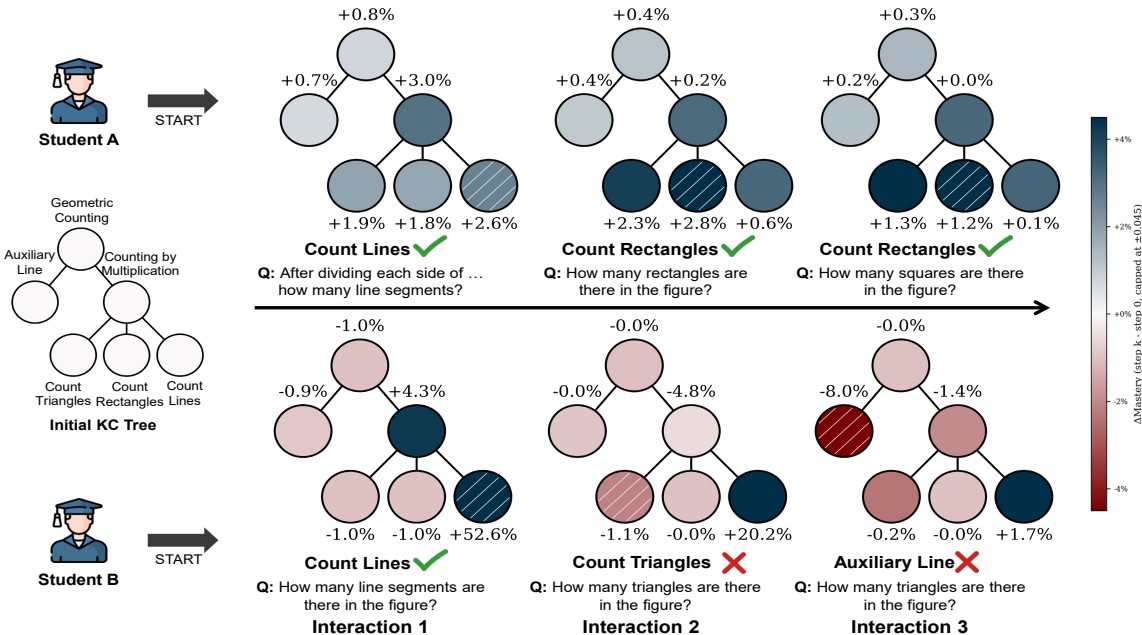

Figure 3: Examples of KC mastery probability update. Node color indicates the cumulative mastery probability change from the initial estimation (after burn-in) to the current step, with blue for increase and red for decrease. The value denotes the step-wise change. The node with dashed lines indicates the current concept. The row under each tree shows the student's interaction at that step.

data compared to $KT^2$, which only performs individual-level updates without using other students' new interactions. Other implementation details are provided in Appendix H

### 4.3   Main Result

Table 2 presents the performance of all methods across both datasets. First, $KT^2$ achieves the highest scores across almost all metrics on both XES3G5M and MOOCRADAR, demonstrating its effectiveness and robustness in the simulated low-resource setting. Even recent methods designed for cold-start (CSKT) and length extrapolation (LEFOKT), are outperformed by $KT^2$ in our low-resource setting, suggesting that specialized architectural designs alone do not suffice under the low-resource regime our method targets.

Second, among the conventional DLKT baselines, all online variants outperform their offline counterparts, highlighting the importance of online updating. Notably, compared with our method, all DLKT-Online baselines have access to more data at each update step, as they retrain using the full classroom's cumulative interactions. In contrast, $KT^2$ updates only with the new interaction from a single student. Despite this apparent advantage, these baselines still underperform, suggesting that simply applying online retraining to existing DLKT models is insufficient to fully capture the evolution of an student's knowledge state.

Third, we also observe that both Qwen and LLaMA do not perform well on both datasets. This suggests that current LLMs may lack the capability to effectively extract student characteristics from historical exercise data and make accurate predictions.

### 4.4   Qualitative Analysis

To better understand how our framework performs real-time updates, we visualize the changes in the posterior mastery probability, $p_\theta(K_{ci} = 1|\mathcal{Q}_i)$, across a KC subtree during incremental inference for two students (Fig. 3), as the historical observation, $\mathcal{Q}_i$, expands. Color reflects differences between the current mastery probability and the initial estimation based on burn-in data only. Each node is annotated with step-wise mastery change. Our findings are as follows:

First, we find the probability updates align with the inputs. In this example, Student A solves all three questions correctly, leading to an increase in mastery probability on all nodes in the tree. Student B answers the first question correctly, but the next two incorrectly. As a result, only the "Count Line" node corresponding to the first question shows a higher mastery probability than the initial estimation, while the rest of the nodes all get lower mastery probabilities.

Second, it is worth noting that due to the tree propagation, the mastery probability changes are not confined to the KCs directly associated with the questions. Other nodes also dynamically adjust their mastery probabilities, even when the KC has not been explicitly visited in the student's exercise history. This effect is evident not only in direct parent or sibling nodes. For example, Student A correctly answering a "Count Lines" question also increases confidence in "Auxiliary Line".

In summary, these findings demonstrate the ability of $KT^2$ to dynamically update and adjust the student's profile in a structured and interpretable way, with respect to the concept hierarchy. We provide a per-student case study illustrating model behaviors in Appendix I.

### 4.5   Ablation Study

Fig. 4ab shows the AUC results on both datasets (average across all modules) under two settings: ❶ different burn-in sizes, and ❷ different classroom sizes. We exclude LLM-based KT methods from these comparison, as their few-shot prompting approach relies on randomly selected examples from the same student rather than leveraging the structured burn-in set and peers' data.

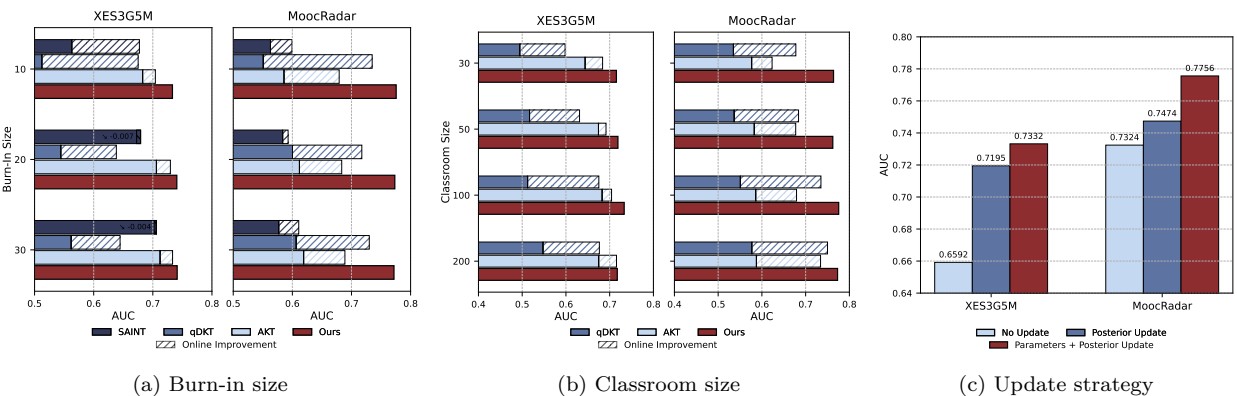

(a) Burn-in size           (b) Classroom size           (c) Update strategy

Figure 4: Ablation studies. (a) Effect of burn-in size. (b) Effect of peers number. (c) Effect of update individualization. Dashed bars/lines denote online improvement; negative gains annotated inline.

As observed, $KT^2$ consistently outperforms the baselines across all burn-in size settings, particularly in low burn-in scenarios, but also remains competitive under larger burn-in sizes. When the burn-in size increases, all offline methods benefit less from online updates. Similar trends are observed when varying classroom size. Our method achieves the best performance in all cases, highlighting its ability to generalize well even with limited peers. The fact that DLKT models require a substantial amount of data to perform well further underscores their limitations in low-resource settings, where the ability to understand students' capabilities from a limited data is crucial.

We further ablate update individualization (Fig. 4c), ranging from fixing both model parameters and student mastery posteriors after burn-in, to updating only the posteriors during inference phrase, and finally to updating both, corresponds to our proposed $KT^2$. Results show that posterior updates alone already improve performance, while full updates consistently achieve the best results.

## 5   Conclusion

In this paper, we propose $KT^2$, a probabilistic framework for low-resource, online knowledge tracing. Leveraging the hierarchical structure of KCs, $KT^2$ builds a Hidden Markov Tree Model, where hidden variables represent concept mastery levels and observed variables correspond to students' correctness on exercise

questions. It enables effective learning from minimal initial data and supports incremental updates as new responses are observed. Experiments across six simulated classroom scenarios show that $KT^2$ consistently outperforms existing baselines. Qualitative analysis further demonstrates the ability of $KT^2$ to dynamically refine mastery estimates and propagate the updates across related concepts. Overall, our work underscores the value of integrating concept hierarchies and probabilistic inference in developing data-efficient, personalized knowledge tracing systems.

## Limitation

Our evaluation is conducted on simulated low-resource settings derived from existing datasets, while real-world deployment remains to be validated. Our model also makes several structural assumptions. First, we model the KC hierarchy as an entailment tree (Sec. 3.2). The entailment relation naturally forms a tree and enables exact, efficient inference via the upward-downward algorithm. Other relation types, such as prerequisite dependencies, can instead form directed acyclic graphs in which a concept has multiple parents. Extending the model to such structures would require approximate inference. Second, each exercise is associated with a single KC (Sec. 3.2). Exercises requiring multiple skills are not directly modeled, and extending the emission model to multi-KC questions is left to future work. Third, the transition model assumes a deterministic parent-to-child mastery relation (Sec. 3.3), i.e., mastering a parent KC entails mastering its children. These assumptions are most appropriate when the KC hierarchy represents true parent-child entailment, and may be less accurate otherwise.

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

## A  Parameter Estimation

In this section, we provide the closed-form solution of $\boldsymbol{\theta}^{(\tau)}$, computed by taking the partial derivative on Eq. equation 13 with respect to each parameter.

$$\gamma_{ci}^{\text{master } \mathcal{P}(c)} = p_{\boldsymbol{\theta}^{(\tau-1)}}(K_{ci} = 1, K_{\mathcal{P}(c)i} = 0 | \mathcal{Q}_i) \tag{15}$$

$$\gamma_{ci}^{\text{not-master } \mathcal{P}(c)} = p_{\boldsymbol{\theta}^{(\tau-1)}}(K_{ci} = 0, K_{\mathcal{P}(c)i} = 0 | \mathcal{Q}_i) \tag{16}$$

$$\gamma_c^{(\tau)} = \frac{\sum_i \gamma_{ci}^{(1)}}{\sum_i \gamma_{ci}^{(0)} + \sum_i \gamma_{ci}^{(1)}}. \tag{17}$$

The closed-form solution of any root node can be obtained similarly by removing the $K_{\mathcal{P}(c)i} = 0$.

We define $\mathcal{N}_{\text{pos}}^{(i)}$ and $\mathcal{N}_{\text{neg}}^{(i)}$ as the sets of correctly and incorrectly answered questions by student $i$, respectively. Then we have

$$\varepsilon_i^{\text{pos}} = \sum_{n:n\in\mathcal{N}_{\text{pos}}^{(i)}} p_{\boldsymbol{\theta}^{(\tau-1)}}(K_{\mathcal{M}(n)i} = 0|\mathcal{Q}_i) \tag{18}$$

$$\varepsilon_i^{\text{neg}} = \sum_{n:n\in\mathcal{N}_{\text{neg}}^{(i)}} p_{\boldsymbol{\theta}^{(\tau-1)}}(K_{\mathcal{M}(n)i} = 0|\mathcal{Q}_i) \tag{19}$$

$$\varepsilon^{(\tau)} = \frac{\sum_i \varepsilon_i^{\text{pos}}}{\sum_i \varepsilon_i^{\text{neg}} + \sum_i \varepsilon_i^{\text{pos}}}. \tag{20}$$

For each difficulty class $l$, where $l \in \{\text{easy}, \text{medium}, \text{hard}\}$, the closed-form solution of the emission probability is

$$r_{l_i}^{\text{pos}} = \sum_{\substack{n:n\in\mathcal{N}_{\text{pos}}^{(i)} \\ \text{difficulty}=l}} p_{\boldsymbol{\theta}^{(\tau-1)}}(K_{\mathcal{M}(n)i} = 1|\mathcal{Q}_i) \tag{21}$$

$$r_{l_i}^{\text{neg}} = \sum_{\substack{n:n\in\mathcal{N}_{\text{neg}}^{(i)} \\ \text{difficulty}=l}} p_{\boldsymbol{\theta}^{(\tau-1)}}(K_{\mathcal{M}(n)i} = 1|\mathcal{Q}_i) \tag{22}$$

$$r_l^{(\tau)} = \frac{\sum_i r_{l_i}^{\text{pos}}}{\sum_i r_{l_i}^{\text{neg}} + \sum_i r_{l_i}^{\text{pos}}}. \tag{23}$$

## B    Upward-Downward Algorithm

For each student $i$, we only need to compute the following posterior distributions, which are sufficient for updating all required parameters.

$$p_{\boldsymbol{\theta}}(K_{\mathcal{M}(n)i} = 1|\mathcal{Q}_i) \tag{24}$$

$$p_{\boldsymbol{\theta}}(K_{\mathcal{M}(n)i} = 0|\mathcal{Q}_i) \tag{25}$$

$$p_{\boldsymbol{\theta}}(K_{ci} = 1, K_{\mathcal{P}(c)i} = 0|\mathcal{Q}_i) \tag{26}$$

$$p_{\boldsymbol{\theta}}(K_{ci} = 0, K_{\mathcal{P}(c)i} = 0|\mathcal{Q}_i) \tag{27}$$

We first introduce two auxiliary probabilities. The first one is **downward probability**

$$\alpha_c(k) = p(K_c = k, \mathcal{Q}_{\mathcal{S}_{\mathcal{V}\backslash\mathcal{D}(c)}} = \boldsymbol{q}_{\mathcal{S}_{\mathcal{V}\backslash\mathcal{D}(c)}}). \tag{28}$$

Here, $\mathcal{D}(c)$ denotes all the descendant nodes of $c$, **including** $c$. $\mathcal{V}\backslash\mathcal{D}(c)$ denotes all the nodes except for $\mathcal{D}(c)$. $\mathcal{Q}_{\mathcal{S}_{\mathcal{V}\backslash\mathcal{D}(c)}}$ represents all the questions that involve concepts in $\mathcal{V}\backslash\mathcal{D}(c)$ and $\mathcal{V}\backslash\mathcal{D}(c)$ only. $\mathcal{S}_c$ denotes the set of questions that involves KC $c$ and that is answered by student $i$ (the subscript $i$ is removed for convenience).

The second probability is called the **upward probability**

$$\beta_c(k) = p(\mathcal{Q}_{\mathcal{S}_{\mathcal{D}(c)}} = \boldsymbol{q}_{\mathcal{S}_{\mathcal{D}(c)}}|K_c = k). \tag{29}$$

The upward probability can be recursively computed from that of the children nodes (denote the children nodes of $c$ as $Child(c)$)

$$\beta_c(k) = \prod_{n\in\mathcal{S}_c} p(Q_n = q_n|K_c = k) \cdot \prod_{j\in Child(c)} \tilde{\beta}_{j,c}(k), \tag{30}$$

where

$$\tilde{\beta}_{j,\mathcal{P}(j)}(k) = p(\mathcal{Q}_{\mathcal{S}_{\mathcal{D}(j)}} = \boldsymbol{q}_{\mathcal{S}_{\mathcal{D}(j)}} | K_{\mathcal{P}(j)} = k) \tag{31}$$

$$= \sum_{k_j} \beta_j(k_j) p(K_j = k_j | K_c = k). \tag{32}$$

At leaf nodes, $\beta_c(k)$ is just a simple emission probability.

The downward probability can be recursively computed from that of the parent node

$$\alpha_c(k) = p(K_c = k, \mathcal{Q}_{\mathcal{S}_{\mathcal{V} \setminus \mathcal{D}(c)}} = \boldsymbol{q}_{\mathcal{S}_{\mathcal{V} \setminus \mathcal{D}(c)}}) \tag{33}$$

$$= \sum_{k_{\mathcal{P}(c)}} p(K_c = k | K_{\mathcal{P}(c)} = k_{\mathcal{P}(c)}) \tilde{\alpha}_{\mathcal{P}(c)}, \tag{34}$$

where

$$\tilde{\alpha}_{\mathcal{P}(c)}(k) = \alpha_{\mathcal{P}(c)}(k_{\mathcal{P}(c)}) \frac{\beta_{\mathcal{P}(c)}(k_{\mathcal{P}(c)})}{\tilde{\beta}_{c,\mathcal{P}(c)}(k_{\mathcal{P}(c)})}. \tag{35}$$

At root nodes, $\alpha_c(k)$ is just the prior distribution.

After all the auxiliary variables are computed, the target posterior distributions can be computed as

$$p_{\boldsymbol{\theta}}(\mathcal{K}_{ci} = 1 | \mathcal{Q}_i) = \frac{\alpha_c(1)\beta_c(1)}{\sum_{k=0}^{1} \alpha_c(k)\beta_c(k)}, \tag{36}$$

$$p_{\boldsymbol{\theta}}(K_{ci} = k_c, K_{\mathcal{P}(c)i} = k_{\mathcal{P}(c)} | \mathcal{Q}_i) = \frac{\tilde{\alpha}_{\mathcal{P}(c)}(k_{\mathcal{P}(c)})\beta_c(k_c)p(k_c | k_{\mathcal{P}(c)})}{\sum_{k'_c, k'_{\mathcal{P}(c)} \in \{0,1\}} \tilde{\alpha}_{\mathcal{P}(c)}(k'_{\mathcal{P}(c)})\beta_c(k'_c)p(k'_c | k'_{\mathcal{P}(c)})}. \tag{37}$$

## C   Use of Large Language Models

In this paper, we utilized LLMs for auxiliary purposes only. Specifically, LLMs were used to polish the writing and correct grammatical issues in the manuscript, and to assist in data translation (see Appendix D) and KC tree construction when only fine-grained KCs are provided (see Appendix F).

## D   Translation

For both XES3G5M and MOOCRADAR, the data translation is done following the pipeline below.

We translated each question into English using GPT-4o-mini with *[Prompt for Translation]*. We specifically required the LLM to convert all fill-in-the-blank questions into a proper question format (e.g., 'There are __ squares in the plot.' should be transformed into 'How many squares are there in the plot?'). Then, the GPT-4o-mini was prompted with *[Prompt for Translation Check]* to self-check the correctness of its translation, considering both the meaning match between the Chinese question and the English translation, as well as the question format conversion.

We noticed that in the XES3G5M dataset, the blank symbol might be missing in some Chinese fill-in-the-blank questions (e.g., the question 'There are __ squares in the plot.' might be recorded as 'There are squares in the plot.'), leading to difficulty in both the translation and the correctness check phases. To handle the incorrect translations, we used a stronger model, GPT-4o, to double-check the translation correctness and generate an explanation for its justification using the same *[Prompt for Translation Check]*. All questions regarded as incorrect by GPT-4o proceeded into an automatic translation revision phase, where GPT-4o was prompted with the *[Prompt for Translation Fix]* to revise the translation utilizing the explanation of why this translation was incorrect.

After one round of correction, GPT-4o checked the correctness of the new translation again. After this stage, the remaining incorrect translations were revised by a human translator and added to the collection of English translations. See Appendix K for all translation prompts.

## E  Datasets

We summarized the statistics of original XES3G5M and MOOCRADAR in Table 3.

**License:** XES3G5M is an open-sourced dataset released under the MIT license, which allow free use of research and educational purposes. MOOCRADAR is also publicly available for academic research. Both datasets have been anonymized to protect student privacy and do not contain personal identifiable information.

| Dataset | Students | Items | KCs | Interactions |
|---------|----------|-------|-----|--------------|
| XES3G5M | 18,066 | 7,652 | 865 | 5.5M |
| MOOCRadar | 14,224 | 2,513 | 5,600 | 12.7M |

Table 3: Dataset statistics.

## F  Knowledge Concept Trees

In this section, we describe the details of knowledge concept trees for the XES3G5M and MOOCRADAR. As noted in Sec. 4.1, XES3G5M provides a predefined hierarchical structure of KCs, which we adopt directly as its knowledge concept tree. In contrast, MOOCRADAR includes only fine-grained KCs without any hierarchical organization, so we construct a knowledge concept tree for it. To begin, we embed the original Chinese KCs using the bge-base-zh model (Xiao et al., 2023), producing semantic embeddings. These embeddings are then reduced in dimensionality using UMAP (McInnes et al., 2018) and clustered with HDBSCAN (Campello et al., 2013) to group semantically similar KCs. This process yields a set of KC clusters along with a number of outliers that do not belong to any cluster. For each cluster, we use GPT-4o-mini to generate a summarized KC label. To maintain robustness, we include an "unsummarizable" option, allowing the model to flag incoherent clusters, which are then manually reviewed and reclassified. For the outlier KCs, we sample a representative exercise from the dataset and prompt GPT-4o-mini again to determine whether the KC can be merged into an existing cluster. If not, it is retained as an independent node and manually labeled on a case-by-case basis.

We also apply post-processing to refine the knowledge concept trees. Since we assume each exercise is associated with a single KC, we retain only the most frequently occurring KC when multiple are assigned. Additionally, we merge child nodes associated with fewer than 10 unique exercises into their parent to avoid sparsity. If only one child remains after merging, its questions are reassigned to the parent, and the child node is pruned.

## G  Module Statistics

Table 4 shows the statistics of all six knowledge modules. The view of each module can be found in Appendix J.

## H  Implementation Details

We initialize our model parameters as in Table 5.

We observe that the guessing probability $\varepsilon$ can exceed 0.3 during training in some cases. To avoid overfitting to spurious correct responses, we clip $\varepsilon$ to a maximum of 0.3 in each EM update iteration. This constraint

| Module | #Nodes | Max Depth | #Leaves |
|---|---|---|---|
| Application Module | 148 | 5 | 100 |
| Computation Module | 95 | 5 | 66 |
| Counting Module | 69 | 5 | 46 |
| Wine Knowledge | 5 | 2 | 4 |
| Circuit Design | 9 | 2 | 8 |
| Education Theory | 4 | 2 | 3 |

Table 4: Statistics of the knowledge modules.

| Parameter | Symbol | Initial Value |
|---|---|---|
| Transition | $\gamma_c$ | 0.1 |
| Emission (easy) | $r_{\text{easy}}$ | 0.9 |
| Emission (medium) | $r_{\text{med}}$ | 0.8 |
| Emission (hard) | $r_{\text{hard}}$ | 0.75 |
| Guessing | $\varepsilon$ | 0.1 |

Table 5: Initialization values for model parameters before EM training.

prevents both overfitting and label-switching, where $K_{ci}$'s interpretation could reverse with $\varepsilon$ inappropriately representing mastery probability rather than guess rate.

For IBKT, we used the open-source pyBKT implementation (Badrinath et al., 2021). For AKT, SAINT, and QDKT, we adopt the standardized implementation provided by PYKT (Liu et al., 2022), a unified Python library for benchmarking KT models. We implement REKT and SKT by their official GitHub repositories. For all LLM baselines, we adopt the publicly released checkpoints.

For all deep-learning-based baseline methods, we follow the default hyperparameters reported in their original papers to train the offline versions. For online variants, we set the learning rate to $1e - 4$ and the batch size to 32, and train for a single epoch at each update step.

For both LLM-based baseline methods, we use the vLLM package (Kwon et al., 2023) for inference, with a sampling temperature of 0.8, a top-p (nucleus) sampling threshold of 0.8, and a repetition penalty of 1.1.

Here we included a runtime comparison of $KT^2$ and other online baselines on representative modules from the two datasets in the table below.

| Model | XES3G5M | MOOCRadar | Device |
|---|---|---|---|
| $KT^2$ | 412.1 s | 224.3 s | CPU |
| AKT-ONLINE | 284.8 s | 196.6 s | RTX A6000 |
| SAINT-ONLINE | 197.5 s | 134.7 s | RTX A6000 |
| QDKT-ONLINE | 74.3 s | 45.4 s | RTX A6000 |
| QWEN-2.5 | 4.5 hours | 3.2 hours | NVIDIA H100 |
| LLAMA-3.2 | 25 min | 16 min | NVIDIA H100 |

Table 6: Runtime comparison across datasets. GPU setup: RTX A6000 (48GB, 1 GPU) and NVIDIA H100 (80GB, 1 GPU).

$KT^2$ requires slightly longer update time, as it performs individualized update for each student, while DLKT-online aggregates interactions across students and retrain less frequently. Despite the additional cost, the method remains efficient and yields consistent performance improvements.

For LLM, we can see that Qwen-2.5-7B takes 4.5h and LLaMA-3.2-3B takes 25 min on a 80G H100 GPU. This suggests that LLM-based inference is significantly more expensive.

## I  Case Study

To better understand when our method works or struggles, we looked into per-student trajectories in our inference logs. Specifically, we examined the three quantities:

1. Predicted correctness $p(Q_{n^*i} = 1 \mid \mathcal{Q}_i)$.

2. KC-level posterior $p(K_{ci} = 1 \mid \mathcal{Q}_i)$ after observing each interaction.

3. The mastery estimation $p(K_{ci} = 1)$, computed through the hierarchical structure as the product of transition probabilities along the path from the root to node $c$:

$$p(K_{ci} = 1) = p(K_{\text{root}\,i} = 1) \prod_{u \in \text{path}(c \to \text{root})} p(K_{ui} = 1 \mid K_{\mathcal{P}(u)i} = 1), \tag{38}$$

where $\text{path}(c \to \text{root})$ denotes the unique ancester chain from concept $c$ to the root.

Overall, we observed that the predicted correctness and KC posterior react quickly to recent answers, while the mastery estimate moves more smoothly, especially for deeper KC nodes influenced by many ancestors. This makes the model stable in early stages, but sometimes not sensitive enough to capture the repeated failures later on.

**A case where $KT^2$ adapts well.** For student 254 on "Addition/Subtraction Problems", two initial mistakes slightly reduce the predicted correctness ($0.496 \to 0.473$). Once the student starts answering correctly, the prediction rises quickly ($0.597 \to 0.705$), while the mastery estimate remains stable around 0.725 throughout the process. This shows rapid adaptation in correctness prediction and a smooth update for mastery estimation.

**A case where $KT^2$ is too conservative.** For student 15302 on "Periodic Arrangement Problems", the hierarchy gives a high mastery estimation ($\sim 0.815$). The student then answers three repeated KC questions incorrectly, but the predicted correctness remains high ($\sim 0.85$), treating these failures as slips. The model only adjusts after several observations, suggesting that strong ancestral influence can delay adaptation.

## J   Module Views

```
Application Module
    +-- Equation Word Problem
    |   +-- Linear Equation Word Problem
    |   \-- Indeterminate Equation Word Problem
    +-- Addition/Subtraction Word Problem
    |   +-- Change in Ratio Word Problem
    |   \-- Simultaneous Increase/Decrease Word Problem
    +-- Period Problem
    |   +-- Basic Arrangement Period Problem
    |   +-- Sequence Operation Period Problem
    |   |   +-- Number Period
    |   |   \-- Find Which Number in Sequence Problem
    |   +-- Period Problem in Time
    |   |   +-- Period in Calendar Dates
    |   |   +-- Period in Time
    |   |   \-- Find Day of the Week for a Date Problem
    |   \-- Cyclic Operation Period Problem
    +-- Sum and Difference Multiple Problem
    |   +-- Basic Calculation of Multiples
    |   |   +-- Two Quantities Multiple
    |   |   \-- Multiple Quantities Multiple
    |   +-- Change Multiple Problem
    |   |   +-- Two Quantities Simultaneously Change Multiple
    |   |   \-- Basic Change Multiple
    |   +-- Sum and Multiple Problem
    |   |   +-- Two Quantities Sum and Multiple Problem
    |   |   |   \-- Two Quantities Sum and Multiple
    |   |   +-- Multiple Quantities Sum and Multiple Problem
    |   |   |   \-- Multiple Quantities Sum and Multiple
    |   |   +-- Find Hidden Sum
    |   |   \-- From Less to More
    |   +-- Sum and Difference Problem
    |   |   +-- Two Quantities Sum and Difference Problem
    |   |   |   +-- Known Differences
    |   |   |   +-- Known and Hidden Difference
    |   |   |   \-- Hidden and Known Difference
    |   |   \-- Multiple Quantities Sum and Difference Problem
    |   \-- Difference Multiple
    |       +-- Two Quantities Difference Multiple
    |       |   +-- Integer Ratio with Difference
    |       |   +-- Hidden Ratio Type Two Quantities Difference Multiple Problem
    |       |   +-- Hidden Difference Type Two Quantities Difference Multiple Problem
    |       |   +-- Non-integer Ratio Difference Multiple Insufficient
    |       |   \-- Non-integer Ratio Difference Multiple Remainder
    |       \-- Multiple Quantities Difference Multiple

    ...

    \-- Chicken-Rabbit Problem
        +-- Assumption Method for Solving Chicken-Rabbit Problem
        |   +-- Back-Deduction Type
        |   +-- Basic Type
        |   |   +-- Prototype Problem
        |   |   \-- Variant Problems
        |   \-- Find Number of Animals
        +-- Grouping Method for Solving Chicken-Rabbit Problem
        |   +-- Head Multiple Type
        |   \-- Flexible Grouping
        +-- Multiple Quantities Chicken-Rabbit Problem
        +-- Using Assumption Method to Solve Modified Chicken-Rabbit Problem
        +-- Using Grouping Method to Solve Modified Chicken-Rabbit Problem
        \-- Using Grouping Method to Solve Basic Chicken-Rabbit Problem
```

```
Computation Module
    +-- Exponentiation
    |   +-- Application of Exponentiation
    |   \-- Exponentiation Operations
    +-- Fraction
    |   +-- Fraction Basics
    |   |   +-- Properties of Fractions
    |   |   \-- Meaning of Fractions
    |   +-- Fraction Tricks
    |   \-- Fraction Operations
    |       +-- Fraction Addition and Subtraction
    |       \-- Addition and Subtraction of Fractions with Different Denominators
    +-- Unit Conversion
    |   +-- Length Unit Conversion
    |   \-- Area Unit Conversion
    +-- Define New Operation
    |   +-- Reverse Solving Unknown Type
    |   \-- Direct Calculation Type
    |       \-- Normal Type
    +-- Decimal
    |   +-- Decimal Addition and Subtraction
    |   |   +-- Decimal Addition and Subtraction Trick with Rounding Method
    |   |   \-- Decimal Addition and Subtraction Vertical Format Calculation
    |   +-- Decimal Four Operations
    |   +-- Decimal Basics
    |   |   +-- Rounding
    |   |   +-- Decimal Comparison
    |   |   +-- Decimal Point Movement Patterns
    |   |   \-- Understanding Decimals
    |   \-- Meaning of Decimals
    |       +-- Decimal Point Movement
    |       \-- Reading and Writing Decimals
    +-- Sequences and Number Tables
    |   +-- Sequence Patterns
    |   +-- Number Table Patterns
    |   |   +-- Finding Patterns by Combining Numbers and Diagrams (Multiple Diagrams)
    |   |   \-- Rectangle Number Table
    |   |       +-- Positional Relationship
    |   |       +-- Find Number at Known Position in Continuous Natural Number Rectangle Table
    |   |       \-- Find Position of Known Number in Continuous Natural Number Rectangle Table
    |   +-- Arithmetic Sequence
    |   |   +-- Application of Mean Value Theorem
    |   |   +-- Truncated Sum of Arithmetic Sequence
    |   |   +-- Find Common Difference of Arithmetic Sequence
    |   |   +-- Sum of Arithmetic Sequence
    |   |   +-- Find General Term of Arithmetic Sequence
    |   |   \-- Find Number of Terms in Arithmetic Sequence
    |   \-- Geometric Sequence

    ...

    +-- Equation Basics
    |   +-- Linear Equation in One Variable
    |   |   \-- Equation with Integer Coefficients
    |   \-- Indeterminate Equation
    +-- Comparison and Estimation
    \-- Induction of Split and General Terms
```

```
Counting Module
    +-- Geometric Counting
    |   +-- Categorical Enumeration of Figures
    |   |   +-- Regular Figure Enumeration Counting
    |   |   |   +-- Categorized Figures
    |   |   |   +-- Square
    |   |   |   +-- Understanding Line Segments
    |   |   |   \-- Rectangle
    |   |   \-- Lattice Point Constructed Figures
    |   +-- Correspondence Method Figures
    |   |   \-- Counting by Multiplication
    |   |       +-- Count Triangles
    |   |       +-- Count Lines
    |   |       \-- Count Rectangles
    |   \-- Auxiliary Line
    +-- Addition-Multiplication Principle
    |   +-- Multiplication Principle
    |   |   +-- Other Types in Multiplication Principle
    |   |   +-- Object Quantity Combination
    |   |   +-- Item Matching
    |   |   +-- Item Matching (With Special Requirements)
    |   |   \-- Route Matching Problem
    |   +-- Comprehensive Addition-Multiplication Principle
    |   +-- Addition Principle
    |   |   +-- Other Types in Addition Principle
    |   |   \-- Handshake and Toasting Problem
    |   +-- Queueing Problem
    |   +-- Coloring Counting Problem
    |   \-- Grouping Problem
    |       +-- General Grouping Problem
    |       \-- Grouping Problem with Special Requirements
    +-- Inclusion-Exclusion Principle
    |   +-- Three-Set Inclusion-Exclusion
    |   +-- Two-Set Inclusion-Exclusion
    |   \-- Geometric Counting with Multi-Set Inclusion-
    ...

    +-- Comprehensive Enumeration Method
    |   +-- Lexicographical Order Method
    |   |   +-- Grouping
    |   |   |   +-- Increasing Numbers (Decreasing Numbers)
    |   |   |   +-- Card Grouping
    |   |   |   +-- Number Grouping (No Repetition)
    |   |   |   +-- Number Grouping (Repetition Allowed)
    |   |   |   \-- Number Grouping (Specified Digit Size)
    |   |   \-- Non-Numeric Permutation
    |   +-- Integer Partition
    |   |   +-- Integer Partition Application
    |   |   |   +-- Multiplicative Partition (Application)
    |   |   |   \-- Additive Partition (Application)
    |   |   \-- Simple Partition
    |   |       +-- Additive Partition (Non-Identical Numbers)
    |   |       \-- Additive Partition (Specified Count)
    |   \-- Enumeration Method
    |       +-- Methods of Payment
    |       +-- Ordered Enumeration
    |       \-- Distribute Items by Given Count
    +-- Counting Method
    |   +-- Standard Counting Method
    |   |   \-- Shortest Path
    |   +-- Special Points or Areas
    |   \-- Stepwise Counting Method
    \-- Statistics and Probability
        +-- Probability
        \-- Statistical Charts
```

```
Wine Knowledge
    +-- Wine
    +-- Wine Evaluation Knowledge
    +-- Wine Aroma Type
    \-- Types of Wine

Circuit Design
    +-- Amplifier Circuit Design
    +-- Ideal Operational Amplifier
    +-- Electronics
    +-- Electrical Concepts
    +-- Electrical Circuit Knowledge
    +-- Circuit Design and Analysis
    +-- Operational Amplifier Knowledge
    \-- Integrated Operational Amplifier Circuit

Education Theory
    +-- Feedback
    +-- Class Teacher Management and Teacher-Student Relationship
    \-- Foundation of Class Formation
```

## K LLM Prompt

In this section, we provide the complete prompts for LLM baselines and dataset translation.

---

**Prompt for LLM Knowledge Tracing:**

Your task is to analyze student's past performance on a series of questions and predict whether students can answer a given question correctly. Below are 10 questions the student has already answered. After each question you will see whether the answer was correct (1) or incorrect (0).

Q1. In a capacitive coupling amplifier circuit, after introducing negative feedback, it is only possible to have low-frequency self-excited oscillation, and high-frequency self-excited oscillation is not possible. Is this statement true or false?
Concepts: Feedback
Correct: 0

Q2. Since some negative feedback amplifiers can produce self-excited oscillations, can they be used as signal sources?
Concepts: Feedback
Correct: 0

Q3. ...
*(Omit the other 8 historical exercises)*

Now consider the new question:
If negative feedback is introduced through a resistor in a single-stage common-emitter amplifier, what will happen?
If negative feedback is introduced through a resistor in a two-stage common-emitter amplifier, what will happen?
A. It will definitely produce high-frequency self-oscillation B. It may produce high-frequency self-oscillation C. It will definitely not produce high-frequency self-oscillation.
Concepts: Feedback
Based on the student's past performance, will the student answer this question correctly? Respond with 1 for correct, 0 for incorrect. Output only 0, 1—no additional text.

Predict:

---

**Prompt for Translation:**
You are a helpful AI assistant skilled in translating Chinese exercises to English.

Guidelines:
- You will be provided with an exercise in Chinese. Your task is to translate it accurately and clearly into English.
- If the Chinese question is a fill-in-the-blank question, convert it into a proper question format in English. Be mindful that the blank symbol might be missing from the original question due to formatting errors.
- Output only the translated English text. Do not include any additional text, explanations, or formatting beyond the translation.

Examples:
User 1:
Here is the math exercise to translate: *(Omit the Chinese question)*
Assistant 1:
Xiao Ming has 10 apples, Xiao Hong has 5 apples, how many more apples does Xiao Ming have than Xiao Hong?
...
*(Omit other 2 examples)*

Here is the math exercise to translate: *(Omit the Chinese question)*

---

**Prompt for Translation Check:**

You are a helpful AI assistant skilled in assessing the quality of Chinese question translations for accuracy and coherence. You will be provided with two versions of an exercise: one in Chinese and one in English. Your task is to evaluate whether the English translation accurately reflects the meaning and intent of the Chinese question and provide a detailed explanation to justify your assessment.
Guidelines:
- If the Chinese question is a fill-in-the-blank question, the English translation must be rewritten into a proper question format without retaining the blank symbol. If it fails to meet this requirement, consider it an incorrect translation.
- Be mindful of potential formatting errors in the Chinese question. For example, the blank symbol in a fill-in-the-blank question might be missing. Carefully discern whether the question is a statement or a fill-in-the-blank question with a missing blank.
- If the Chinese question references an image, it is acceptable for the English translation to omit the image filename as long as it maintains the question's meaning.
- Output your evaluation in JSON format using the provided template. First, explain why the English translation is correct or incorrect, then provide a final justification as a boolean value (True for correct, False for incorrect).
- Do not include any additional text or explanations beyond the required JSON output.

Template:
{
"explanation": (Your explanation on why you think the English translation is correct/wrong),
"correct_translation": (a boolean value, True if the English translation is a correct translation, False otherwise)
}

Examples:
User 1:
Here is the Chinese question:*(Omit the Chinese question)*
Here is the English translation of the question: A number, when divided by 4, added 4, multiplied by 4, and then subtracted 4, results in 16. What is the number?
Assistant 1:
{
"explanation": "The Chinese question describes a sequence of mathematical operations performed on a number, leading to the result of 16. The English translation accurately conveys the meaning of the Chinese question by describing the same sequence of operations and the expected result, while appropriately converting the blank into the question format 'What is the number?'. Both the meaning and format are correct.",
"correct_translation":
}
...
*(Omit other 2 examples)*

Here is the Chinese question:*(Omit the Chinese question)*
Here is the English translation of the question:*(Omit the English translation)*

**Prompt for Translation Fix:**

You are a helpful AI assistant skilled in improving the translation of Chinese questions to ensure accuracy and coherence. You will be provided with two versions of a question: one in Chinese and one in English, along with an explanation of why the translation is incorrect. Your task is to rewrite the English translation based on the given explanation to make it correct and consistent with the original Chinese question.

Guidelines:
- Be mindful of potential formatting errors in the Chinese question. For example, the blank symbol in a fill-in-the-blank question may be missing. Carefully discern whether the question is a statement or a fill-in-the-blank question with a missing blank.
- If the Chinese question is a fill-in-the-blank question, rewrite the English translation as a proper question without retaining the blank symbol.
- Provide only the corrected English translation as the output. Avoid including additional explanations or text.

Examples:
User 1:
Here is the Chinese question:*(Omit the Chinese question)*
Here is the English translation you should rewrite: A number, when divided by 4, added 4, multiplied by 4, and then subtracted 4, results in 16. Then the number is ()
The reason why the translation is incorrect: The Chinese question is a fill-in-the-blank question as indicated by the blank symbol (), which requires the English translation to be reformatted into a proper question format without the blank. The provided English translation retains the blank, which does not conform to the specified criteria for a correct translation.
Assistant 1:
Xiao Ming has 10 apples, Xiao Hong has 5 apples, how many more apples does Xiao Ming have than Xiao Hong?

...
*(Omit other 2 examples)*

Here is the math exercise to translate: *(Omit the Chinese question)*
Here is the English translation you should rewrite:*(Omit the English translation)*
The reason why the translation is incorrect: *(Omit the explanation)*

