# OpenReview forum: "A Hierarchical Probabilistic Framework for Incremental Knowledge Tracing in Classroom Settings"
_TMLR — Under review for TMLR_

### Review · Reviewer_kwkS · 2026-06-15

**Summary Of Contributions:**

The paper proposes KT², a hierarchical probabilistic framework for low-resource and online knowledge tracing. It uses a tree-structured hierarchy of knowledge concepts as a structural prior, models each student’s mastery of each concept as a latent variable, and predicts exercise correctness through a Hidden Markov Tree model. The model is estimated with EM and incrementally updated as new student responses arrive.

The main contributions are: (1) formulating a practical KT setting with cold start, limited peers, and streaming interactions; (2) introducing an interpretable knowledge-tree-based probabilistic model for student mastery estimation; and (3) showing that KT² outperforms traditional, deep learning, online, and LLM-based baselines on simulated classroom settings from XES3G5M and MOOCRadar.

The paper’s strengths are its practical motivation, simple and interpretable design, and efficient online personalization. Its main weaknesses are the strong assumptions of a tree-structured concept hierarchy, single-concept question labels, and deterministic parent-child mastery relations. The evaluation is also based on simulated classroom subsets rather than real classroom deployment, and stronger comparisons with hierarchical or Bayesian KT baselines would make the empirical claims more convincing.

**Audience:**

Yes

**Audience Explanation:**

The paper studies a practical setting that is often underemphasized in standard KT benchmarks: predicting student performance when there are few students, limited historical responses, and a need for online updates.

The main finding—that hierarchical knowledge concept structures can provide useful priors for low-resource and incremental knowledge tracing—is potentially valuable. The proposed framework is also interpretable and computationally simple, which may appeal to readers interested in practical and deployable student modeling systems.

That said, the paper may be of narrower interest to the broader TMLR audience because the technical novelty is moderate and the evaluation is limited to simulated classroom settings. Its strongest appeal is likely to researchers interested in structured probabilistic models, educational AI, and data-efficient personalization rather than the general machine learning community.

**Broader Impact Concerns:**

The paper does not appear to raise severe immediate ethical risks, but I think a Broader Impact Statement would be useful given the educational application. Knowledge tracing models can influence how students are assessed, grouped, or assigned learning materials, so inaccurate mastery estimates may unfairly disadvantage some students, especially in low-resource settings where individual histories are sparse. The paper should discuss the risk of over-reliance on automated predictions and emphasize that such systems should support, rather than replace, teacher judgment.

**Claims And Evidence:**

Yes

**Claims Explanation:**

The main claims are mostly supported by the presented evidence, but some of them should be stated more cautiously. The experimental results clearly show that KT² outperforms a broad set of baselines on the constructed low-resource online settings, and the ablation studies support the usefulness of incremental updating and personalization. The qualitative visualization also provides intuitive evidence that the model can propagate mastery updates over the knowledge concept tree.

However, the evidence is less convincing for the broader claim that the method is effective in realistic classroom settings. The experiments are based on simulated classroom subsets sampled from existing datasets rather than real classroom deployments. In addition, the method relies on strong assumptions, such as tree-structured concepts, single-concept question labels, and deterministic parent-to-child mastery relations, but the paper does not provide enough empirical validation of these assumptions. The constructed hierarchy for MOOCRadar also needs stronger validation.

Overall, the empirical evidence supports the paper’s narrower claim that KT² performs well under the authors’ simulated low-resource online KT setup. The evidence is clear and generally convincing, but not sufficient to fully justify broader real-world classroom claims without additional validation.

**Requested Changes:**

1. Clarify and justify the modeling assumptions. The paper should provide stronger justification for the tree-structured KC hierarchy, the single-KC-per-question assumption, and especially the deterministic parent-to-child mastery relation. At minimum, the authors should discuss when these assumptions are expected to hold and when they may fail.
2. Add stronger validation of the constructed MOOCRadar hierarchy. Since the hierarchy is a central input to the proposed method, the paper should report how reliable this constructed tree is, for example through expert validation, sensitivity analysis, or comparison with alternative hierarchy construction methods.
3. Temper the real-world classroom claims. The experiments are conducted on simulated classroom subsets rather than actual classroom deployments. The paper should make this distinction clearer and avoid overclaiming practical effectiveness in real classrooms without deployment evidence.

---

> ### Author Response · Authors · 2026-06-27
> **Response to reviewer kwkS's comments**
>
> We thank reviewer kwks for your valuable comments.
>
> **Q1: Clarify and justify the modeling assumptions.**
>
> We thank the reviewer. The three assumptions are addressed as follows, and we have made the relevant discussion explicit in the revised paper.
>
> Tree structure. Our model is built on an entailment hierarchy, where a parent KC subsumes its children. Under this entailment semantics, the structure is naturally a tree, since a finer concept is contained in a single broader concept rather than depending on several. This is what makes the upward-downward algorithm and our closed-form EM updates applicable. Extending to other relation types is less straightforward. For example,  prerequisite relations are also common but naturally form DAGs (a concept may have multiple prerequisites), where a node can have multiple parents and exact tree-based inference no longer applies.
>
> Single KC per question. We assume each question maps to one leaf KC. This holds well for the datasets we study, where exercises are annotated at a fine-grained concept level, but may fail for exercises that genuinely require composing multiple skills.
>
> Deterministic parent-to-child relation. Mastering a parent KC is assumed to entail mastering its children (Eq. 7). This is a reasonable approximation when child concepts are true sub-skills of the parent, but may not hold when a child is only loosely dependent on its parent.
>
> We have added a Limitations section covering the single-KC and parent-to-child assumptions and when they may fail (p.11, in blue).
>
>
> **Q2: Add stronger validation of the constructed MOOCRadar hierarchy.**
>
> We thank the reviewer. Since the MOOCRadar hierarchy is automatically constructed and central to our method, we agree it is worth examining its reliability, and we clarify what level of quality our method actually requires.
>
> We included MOOCRadar to test whether our method remains usable when a dataset provides no human-annotated KC hierarchy. The tree is built fully automatically without human curation. Our aim is not a perfect hierarchy but one that is reasonable enough for the method to work, and the fact that KT² achieves the best performance on MOOCRadar already indicates the automatically built tree is adequate for this purpose.
>
> To directly assess its reliability, we conducted a small-scale human annotation, where two annotators independently judged the grouping nodes within the three modules used in our experiments on their coherence: whether the KCs merged under a parent node genuinely belong together. Coherence is high, with 77% of nodes fully coherent and 88% coherent or borderline, and inter-annotator agreement (how often the two judges returned the same verdict) of 77%. The fewl coherence errors arise where KC names are lexically similar and easily conflated by the LLM, for example "electron" (in physics) and "electronics" (in electrical engineering). This supports our central point that even without a pre-existing human-annotated hierarchy, a reasonable automatically constructed tree suffices for our method to remain effective.
>
>
> **Q3: Temper the real-world classroom claims.**
>
> We thank the reviewer for pointing this out. We have tightened our claims throughout to refer to the simulated low-resource setting instead of real classroom deployment, and we now state explicitly that our experiments are conducted on simulated classroom subsets rather than actual deployments. We have also added a Limitations section discussing the gap between these simulated testbeds and real classroom deployment. These changes appear in the revised Abstract, Introduction, and Conclusion, and in the new Limitations section (p.11, in blue).

---

> > ### Comment · Reviewer_kwkS · 2026-07-18
> > **Response to rebuttal**
> >
> > Thank you for the clarification. My concerns have been fully addressed.

---

### Review · Reviewer_w92A · 2026-06-17

**Summary Of Contributions:**

This paper proposes $KT^2$, a probabilistic framework for knowledge tracing (KT) designed for low-resource and online classroom settings. The core idea is to model a student's evolving knowledge state using a Hidden Markov Tree Model, leveraging the hierarchical structure of knowledge concepts. The model parameters are initialized via the EM algorithm and updated incrementally. The authors evaluate their method on artificially constructed low-resource subsets of the XES3G5M and MOOCRadar datasets, claiming that their model outperforms several existing KT baselines.

**Audience:**

Yes

**Audience Explanation:**

Yes. KT remains a fundamental and highly relevant topic within the domain of intelligent education.

**Claims And Evidence:**

No

**Claims Explanation:**

While the proposed methodology itself appears conceptually reasonable, the experimental validation and comparison with existing literature are currently insufficient to provide convincing evidence for the paper's claims.

My primary concerns regarding the evidence are twofold:

1. The paper lacks a thorough survey of recent advancements in the KT domain. Several key contemporary papers, particularly those addressing structure-aware KT[1,2,3] and specifically targeting cold-start scenarios[4,5,6] are missing. Consequently, the selected baselines are predominantly from around 2020, which are out-of-date. To convincingly demonstrate the superiority of the proposed method, it is crucial to benchmark it against more recent, state-of-the-art models.

[1] Liu Q, Huang Z, Yin Y, et al. Ekt: Exercise-aware knowledge tracing for student performance prediction[J]. IEEE Transactions on Knowledge and Data Engineering, 2019, 33(1): 100-115.

[2] Chen P, Lu Y, Zheng V W, et al. Prerequisite-driven deep knowledge tracing[C]//2018 IEEE international conference on data mining (ICDM). IEEE, 2018: 39-48.

[3] Li Q, Huang Z, Sun J, et al. HKT: Hierarchical structure-based knowledge tracing[J]. Information Processing & Management, 2025, 62(5): 104206.

[4] Eglington L. Evolutionary Features for Mitigating Cold Starts in Logistic Knowledge Tracing[C]//Proceedings of the 18th International Conference on Educational Data Mining. 2025: 403-409.

[5] Guo Y, Shen S, Liu Q, et al. Mitigating cold-start problems in knowledge tracing with large language models: an attribute-aware approach[C]//Proceedings of the 33rd ACM International Conference on Information and Knowledge Management. 2024: 727-736.

[6] Jung H, Yoo J, Yoon Y, et al. CLST: Cold-Start Mitigation in Knowledge Tracing by Aligning a Generative Language Model as a Students’ Knowledge Tracer[J]. Journal of Educational Data Mining, 2025, 17(2): 86-117.

2. The authors do not adequately explain the rationale behind selecting the current two datasets for evaluation. Since the paper targets cold-start and low-resource settings, the evaluation would be much more rigorous if it included datasets that are naturally sparser and widely recognized in the community for cold-start evaluations (e.g., data from NeurIPS 2020 Education Challenge). Relying only on the current datasets, especially with artificial sampling, limits the generalizability of the claims.

**Requested Changes:**

1. The authors need to expand the literature review to include a more thorough and up-to-date survey of KT.

2. The current deep learning baselines evaluated in the paper (such as AKT, SAINT, and QDKT) are predominantly from 2020. To properly substantiate the performance claims of the proposed $KT^2$ framework, the authors need to incorporate more recent, state-of-the-art KT models into their comparative analysis.

3. The current experimental setup relies on artificially sampling and filtering the XES3G5M and MOOCRadar datasets to simulate a low-resource setting. This artificial construction raises concerns regarding the generalizability of the results to real-world scenarios.

---

> ### Author Response · Authors · 2026-06-27
> **Response to reviewer w92A's comments**
>
> We thank reviewer w92A for your valuable comments.
>
> **Q1: The authors need to expand the literature review to include a more thorough and up-to-date survey of KT.**
>
> We thank the reviewer for the helpful literature suggestions. We have carefully studied the recommended works and found them highly relevant to our direction, especially regarding structure-aware KT and cold-start KT. Accordingly, we have expanded the literature review and incorporated these additional works into the related work discussion. The newly added text is marked in blue in the revised paper.
>
> **Q2: To properly substantiate the performance claims of the proposed  framework, the authors need to incorporate more recent, state-of-the-art KT models into their comparative analysis.**
>
> We thank the reviewer for the suggestion. Among the four recent methods cited by the reviewer, we were unable to find the official release codes for HKT [3], Evolutionary Features [4], and CLST [6], so faithful reproduction within the revision window is not feasible. EAKT [5] provides a repository, but it does not include code for its core contribution, i.e., the LLM-based generation of per-question attributes, and it supplies only the finished attributes for the original 50-question dataset. The implementation is also hardcoded to that dataset's size and schema. Reproducing it on our datasets would require re-implementing the method in full rather than running it as a baseline, which is not feasible within the revision window. To compare with recent methods, we have added two 2025 baselines with runnable implementations in pyKT: csKT (Bai et al., 2025a) and LefoKT (Bai et al., 2025b). These methods are purpose-built for cold-start and length extrapolation respectively, closely aligned with the low-resource setting we target. Both are reported in our revised paper (p.8-9, in blue). KT$^2$ outperforms both across all six modules.  AUC on all six modules (full results in updated Table 2):
>
> | Method | App. | Comp. | Count. | Wine | Circuit | Edu. |
> |------------|--------|--------|--------|--------|---------|--------|
> | csKT | 0.6694 | 0.6850 | 0.6882 | 0.5849 | 0.5454 | 0.6317 |
> | LefoKT | 0.6595 | 0.6756 | 0.5997 | 0.5620 | 0.5497 | 0.6252 |
> | KT^2| 0.7448 | 0.7079 | 0.7470 | 0.7714 | 0.7662 | 0.7891 |
>
> **Q3: The current experimental setup relies on artificially sampling and filtering the XES3G5M and MOOCRadar datasets to simulate a low-resource setting. This artificial construction raises concerns regarding the generalizability of the results to real-world scenarios.**
>
> We thank the reviewer and clarify both the dataset choice and the scope of our claims.
>
> Dataset selection. Our method requires a hierarchical KC structure, which most KT datasets do not provide. We chose XES3G5M and MOOCRadar to cover two complementary conditions under this constraint: XES3G5M (K-12 math) provides a native human-curated hierarchy, while MOOCRadar (university-level, diverse courses, fine-grained KCs) provides none. Since most real datasets likewise lack one, we include it specifically to test whether our method remains effective when the tree is constructed automatically, without human curation. This directly probes whether the method generalizes beyond datasets that happen to come with a curated hierarchy.
>
> On sampling and claim scope. The down-sampling is a controlled construction used to create low-resource settings with known resource levels. Our ablations (Fig. 4) vary burn-in and classroom sizes precisely to check robustness across these levels. That said, we agree our evaluation is on simulated low-resource subsets rather than real classroom deployments. We have accordingly tightened our claims throughout to refer to the simulated low-resource setting, and avoid asserting effectiveness in real-world classroom deployment without deployment evidence (see revised Abstract/Intro/Conclusion, in blue).
>
> On a naturally sparse dataset. We agree that evaluating on a naturally sparse, widely-used dataset would strengthen the generalizability of our claims. We are currently working on incorporating the Eedi (NeurIPS 2020 Education Challenge) data, which provides a subject-level hierarchy compatible with our method's requirements. This requires preprocessing the dataset from scratch into our protocol and re-running baselines under our setting, which is engineering-intensive. We are actively pursuing this and will report results before the discussion deadline.

---

> ### Author Response · Authors · 2026-06-30
> **Follow-up on Q3: additional results on Eedi**
>
> As promised in our earlier response, we have completed the additional evaluation on Eedi (NeurIPS 2020 Education Challenge, Task 1), which is a naturally sparse, widely-used dataset that provides a human-labeled KC hierarchy. Following the preprocessing pipeline described in the paper, we evaluate on the two largest Level-1 modules (the direct subtrees of the root KC node "Maths"), selected by total number of interactions: Number (7.4M) and Algebra (3.5M). For each module we simulate a classroom of 100 students. The Number module spans 89 KCs and 12,018 distinct questions, with students answering 158 interactions on average (median 145); Algebra spans 78 KCs and 5,656 questions, with 140 interactions on average (median 116). This yields 14,820 and 13,041 evaluated interactions respectively.
>
> We compare against AKT, AKT-Online, csKT, and LefoKT, using the same baselines and online-update setup as in our main experiments. As described there, AKT-Online retrains on the whole classroom's cumulative interactions at each step and therefore sees considerably more data than KT^2. KT^2 attains the best AUC and ACC on both modules, ahead of every baseline.
>
> | Model        | Number AUC | Number ACC | Algebra AUC | Algebra ACC |
> |--------------|------------|------------|-------------|-------------|
> | KT^2   | 0.7107 | 0.6825 | 0.7056  | 0.6700  |
> | AKT          | 0.6538     | 0.6544     | 0.5693      | 0.6329      |
> | AKT-Online   | 0.6599     | 0.6553     | 0.6693      | 0.6582      |
> | csKT         | 0.6423     | 0.6529     | 0.5929      | 0.6358      |
> | LefoKT       | 0.6190     | 0.6457     | 0.6215      | 0.6329      |

---

> > ### Comment · Reviewer_w92A · 2026-07-01
> >
> > Many thanks to the authors for their comprehensive replies. Their replies and revisions have thoroughly addressed all of my concerns.

---

### Review · Reviewer_icoc · 2026-06-18

**Summary Of Contributions:**

This paper proposes $KT^2$, a hierarchical probabilistic framework for knowledge tracing in realistic classroom settings characterized by limited peer data, cold-start students, and online updates. The key idea is to exploit hierarchical knowledge concept (KC) structures through a Hidden Markov Tree Model, where latent variables represent student mastery over concepts and observed variables correspond to exercise responses. The framework performs an initial class-level estimation using a burn-in dataset and subsequently updates personalized student models through incremental EM inference.

The paper makes three main contributions:

1.It formulates knowledge tracing as a hierarchical probabilistic inference problem over a knowledge concept tree rather than relying on latent neural representations.

2.It introduces an efficient incremental update mechanism that supports online personalization with limited data.

3.It constructs realistic low-resource classroom evaluation settings and demonstrates strong performance against probabilistic, deep learning, and LLM-based baselines.

Strengths:
1. Strong motivation targeting practical classroom scenarios.
2. Effective use of hierarchical concept structures.
3. Consistent empirical improvements across multiple datasets and evaluation settings.
4. Clear qualitative analysis illustrating concept-level updates.

Weaknesses:
1. The experimental setting is largely simulated and may not fully reflect real deployment environments.
2. Some design choices in the probabilistic model appear heuristic and require stronger justification.
3. The paper could provide more analysis on the robustness of $KT^2$ to noisy or imperfect KC trees.

**Audience:**

Yes

**Audience Explanation:**

Knowledge tracing remains an active research area at the intersection of machine learning, educational data mining, probabilistic modeling, and intelligent tutoring systems. The paper addresses an important gap between benchmark-oriented KT research and realistic classroom deployment constraints.

**Broader Impact Concerns:**

No.

**Claims And Evidence:**

Yes

**Claims Explanation:**

Knowledge tracing remains an active research area at the intersection of machine learning, educational data mining, probabilistic modeling, and intelligent tutoring systems. The paper addresses an important gap between benchmark-oriented KT research and realistic classroom deployment constraints.

1.The evidence would be stronger if the authors discussed the simulation-to-deployment gap more concretely. Real classroom deployment may involve noisy KC labels, multi-concept questions, non-tree prerequisite structures, missing responses, irregular student participation, instructional interventions between interactions, and feedback loops caused by adaptive recommendations.

2.Some modeling assumptions also need stronger justification. For example, Eq. (7) assumes that mastery of a parent concept implies mastery of all child concepts with probability one. Such deterministic hierarchical propagation may not always hold in practice, especially for imperfect or loosely defined educational taxonomies. More empirical justification or sensitivity analysis would strengthen confidence in the model design.

3.The baseline coverage is reasonable but not fully sufficient. The authors should discuss or compare with recent related works such as Towards LLM-Empowered Knowledge Tracing via LLM-Student Hierarchical Behavior Alignment in Hyperbolic Space and Interpretable Knowledge Tracing with Multiscale State Representation, since they are closer to this paper in terms of hierarchical, semantic, hyperbolic, or multiscale student modeling.

**Requested Changes:**

1.It would be helpful to discuss the gap between the simulated classroom testbeds and real classroom deployment. Real classrooms may involve noisy KC annotations, multi-skill exercises, irregular participation, missing responses, and instructional interventions between observed interactions.

2.The baseline comparison could be strengthened by including or discussing recent closely related KT methods. In particular, the authors may consider comparing with or discussing hierarchical and semantic KT methods such as LLM-student hierarchical behavior alignment in hyperbolic space and multiscale interpretable KT.

3.The paper could benefit from a sensitivity analysis on the quality of the KC tree. Since the proposed method relies heavily on the hierarchy, noisy, incomplete, or automatically constructed trees may affect the model’s performance.

4.It would be useful to discuss whether $KT^2$ can be extended beyond tree structures. Many real educational concept structures may be closer to DAGs or general graphs than strict trees.

---

> ### Author Response · Authors · 2026-06-27
> **Response to reviewer icoc's comments - 1**
>
> We thank reviewer icoc for your valuable comments.
>
> **Q1: Discuss the gap between the simulated classroom testbeds and real classroom deployment.**
> We thank the reviewer for raising this. Our evaluation is conducted on simulated low-resource subsets of XES3G5M and MOOCRadar, which do differ from real classroom deployment in important ways. Some of the differences the reviewer mentions are properties of real classrooms that our simulated setup does not reproduce, such as noisy KC annotations, irregular participation, and learning that happens between logged interactions. We acknowledge these as part of the simulation-to-deployment gap rather than something our current experiments address. Two other points connect directly to our modeling choices. We assume a single KC per question, so exercises that genuinely require multiple skills are not modeled directly. We also treat the KC hierarchy as given, which is why hierarchy quality matters.
>
> We have added a Limitations section covering these points in our revised paper (p.11, in blue), and have tightened our claims throughout to refer to the simulated low-resource setting rather than real classroom deployment (see revised Abstract/Intro/Conclusion, in blue).
>
>
>
> **Q2: The baseline comparison could be strengthened by including or discussing recent closely related KT methods.**
>
> We thank the reviewer for these pointers. For the two specific methods mentioned, we briefly discuss the comparison here, and have added these methods to the related work (in blue).
>
> L-HAKT (Fu et al., 2026) also targets hierarchical concept structure, but reaches it very differently. It deploys two LLM agents, a teacher agent that parses question semantics to build parent–child knowledge dependencies, and a student agent that simulates learning behavior to generate synthetic interactions. It then encodes the hierarchy geometrically by optimizing curvature in hyperbolic space. KT² instead is a lightweight probabilistic model that operates directly on the KC tree with closed-form EM updates, designed for the low-resource, online setting rather than relying on LLM-generated data or hyperbolic training.
>
> MIKT (Sun et al., 2024) traces knowledge states at coarse and fine granularities to balance interpretability and performance. It captures multiscale state representations rather than an explicit KC hierarchy, and follows a deep sequential modeling approach distinct from our probabilistic tree-based formulation.
>
>
> **Q3: The paper could benefit from a sensitivity analysis on the quality of the KC tree. Since the proposed method relies heavily on the hierarchy, noisy, incomplete, or automatically constructed trees may affect the model’s performance.**
>
> We thank the reviewer for raising this. We would like to clarify the role the KC tree plays in our MOOCRadar experiments, which we think addresses the underlying concern.
>
> XES3G5M provides high-quality human-curated KC hierarchy, but most datasets do not. We included MOOCRadar specifically to answer a practical question: when no human-annotated hierarchy is available, is our method still usable with an automatically constructed tree that is simple to set up? The MOOCRadar tree is therefore built by a fully automatic pipeline with no human curation. This means our reported MOOCRadar results are already obtained on a non-idealized, automatically constructed tree, and KT² still achieves the best performance. In other words, the method does not require a perfectly constructed hierarchy to work. A reasonable automatically built tree is sufficient.
>
> To assess the reliability of the automatic tree-building pipeline, we additionally conducted a small-scale human annotation, where two annotators independently judged the grouping nodes within the three modules used in our experiments on their coherence: whether the KCs merged under a parent node genuinely belong together. Coherence is high, with 77% of nodes fully coherent and 88% coherent or borderline, and inter-annotator agreement (how often the two judges returned the same verdict) of 77%.  The few coherence errors arise where KC names are lexically similar and easily conflated by the LLM, for example "electron" (in physics) and "electronics" (in electrical engineering).
>
> We note that our goal is not to produce a high-quality hierarchy, but one that is reasonable enough for the method to remain effective, which is precisely the question MOOCRadar is meant to test.

---

> ### Author Response · Authors · 2026-06-27
> **Response to reviewer icoc's comments - 2**
>
> **Q4: It would be useful to discuss whether can be extended beyond tree structures. Many real educational concept structures may be closer to DAGs or general graphs than strict trees.**
>
> We thank the reviewer for raising this. Our method is built on an entailment hierarchy, where a parent KC entails its children. Under this entailment semantics the structure is naturally a tree, since a finer concept is contained in a single broader concept. So within our setting the tree is not an approximation of a more general graph, but the natural form of the entailment relation, and it is what makes exact inference via the upward-downward algorithm and our closed-form EM updates possible.
>
> The reviewer is right that other relation types are closer to DAGs or general graphs. Prerequisite relations are a good example, where a concept may have several prerequisites, so a node can have multiple parents. Extending our model to such relations is a meaningful direction, but it is not a trivial change of input. Once a node has multiple parents, the upward-downward algorithm no longer applies, and exact inference is replaced by approximate methods, trading off some of the efficiency and exactness. We see this as a promising avenue for future work.

---

> > ### Comment · Reviewer_icoc · 2026-07-03
> > **Response to rebuttal**
> >
> > Thanks for authors' reply. My concerns have already been solved.